# LogiCity: Advancing Neuro-Symbolic AI with Abstract Urban Simulation

**Bowen Li**[1]    **Zhaoyu Li**[2]    **Qiwei Du**[3]    **Jinqi Luo**[4]    **Wenshan Wang**[1]    **Yaqi Xie**[1]
**Simon Stepputtis**[1]    **Chen Wang**[3]    **Katia Sycara**[1]    **Pradeep Ravikumar**[1]
**Alexander Gray**[5]    **Xujie Si**[2,6]    **Sebastian Scherer**[1] *

[1]Carnegie Mellon University    [2]University of Toronto    [3]University at Buffalo
[4]University of Pennsylvania    [5]Centaur AI Institute    [6]CIFAR AI Chair, Mila
{bowenli2, basti}@andrew.cmu.edu

## Abstract

Recent years have witnessed the rapid development of Neuro-Symbolic (NeSy) AI systems, which integrate symbolic reasoning into deep neural networks. However, most of the existing benchmarks for NeSy AI fail to provide long-horizon reasoning tasks with complex multi-agent interactions. Furthermore, they are usually constrained by fixed and simplistic logical rules over limited entities, making them far from real-world complexities. To address these crucial gaps, we introduce LogiCity, the first simulator based on customizable first-order logic (FOL) for an urban-like environment with multiple dynamic agents. LogiCity models diverse urban elements using semantic and spatial *concepts*, such as $\texttt{IsAmbulance(X)}$ and $\texttt{IsClose(X, Y)}$. These concepts are used to define FOL rules that govern the behavior of various agents. Since the concepts and rules are *abstractions*, they can be universally applied to cities with any agent compositions, facilitating the instantiation of diverse scenarios. Besides, a key feature of LogiCity is its support for user-configurable abstractions, enabling customizable simulation complexities for logical reasoning. To explore various aspects of NeSy AI, LogiCity introduces two tasks, one features long-horizon sequential decision-making, and the other focuses on one-step visual reasoning, varying in difficulty and agent behaviors. Our extensive evaluation reveals the advantage of NeSy frameworks in *abstract* reasoning. Moreover, we highlight the significant challenges of handling more complex abstractions in long-horizon multi-agent scenarios or under high-dimensional, imbalanced data. With its flexible design, various features, and newly raised challenges, we believe LogiCity represents a pivotal step forward in advancing the next generation of NeSy AI. All the code and data are open-sourced at our website.

## 1   Introduction

Unlike most existing deep neural networks [1, 2], humans are not making predictions and decisions in a relatively black-box way [3]. Instead, when we learn to drive a vehicle, play sports, or solve math problems, we naturally leverage and explore the underlying symbolic representations and structure [3–5]. Such capability enables us to swiftly and robustly reason over complex situations and to adapt to new scenarios. To emulate human-like learning and reasoning, the Neuro-Symbolic (NeSy) AI community [6] has introduced various hybrid systems [7–19], integrating symbolic reasoning into deep neural networks to achieve higher data efficiency, interpretability, and robustness[1].

Despite their rapid advancement, many NeSy AI systems are designed and tested only in very simplified and limited environments, such as visual sudoku [20], handwritten formula recogni-

---

[1]This work mainly focuses on logical reasoning within the broad NeSy community.

38th Conference on Neural Information Processing Systems (NeurIPS 2024) Track on Datasets and Benchmarks.

tion [13], knowledge graphs [21], and reasoning games/simulations [7, 22–26] (see Table 1). A benefit of such environments is that they usually provide data with symbolic annotations, which the NeSy AI systems can easily integrate. However, they are still far from real-world complexity due to the lack of three key features: (1) Most simulators are governed by propositional rules tied to specific fixed entities [13, 20, 23] rather than *abstractions* [7]. As a result, agents learned from them are hard to generalize compositionally. (2) In real life, we learn to reason gradually from simple to complex scenarios, requiring the rules within the environment to be *flexible*. Either overly simplified [7, 20, 24] or overly complicated/unsuitable [27, 28] environments cannot promote the development of NeSy AI systems. (3) Few simulators offer realistic *multi-agent* interactions, where the environment agents often need to actively adapt their behaviors in response to varying actions of the ego agent. Moreover, a comprehensive benchmark needs to provide both *long-horizon* (e.g., > 20 steps) [7] and *visual reasoning* [20] scenarios to exercise different aspects of NeSy AI.

To address these issues, we introduce LogiCity, the first customizable first-order-logic (FOL)-based [29] simulator and benchmark motivated by complex urban dynamics. As illustrated in Figure 1, LogiCity allows users to freely customize spatial and semantic conceptual attributes (concepts), FOL rules, and agent sets as configurations. Since the concepts and rules are *abstractions*, they can be universally applied to any agent compositions across different cities. For example, in Figure 1, concepts such as "IsClose(X, Y), IsAmbulance(Y)", and rules like "Stop(X):-IsAmbulance(Y), IsClose(X, Y)" can be *grounded* with specific and

Table 1: Comparison of existing NeSy benchmarks and LogiCity. Our simulator is governed by diverse *abstractions*, which can be flexibly customized. We also support long-horizon, multi-agent tasks and RGB rendering. "−" denotes partially supported features.

| Benchmarks \ Features | Abstract | Flexible | Multi-Agent | Long-Horizon | RGB |
|---|---|---|---|---|---|
| Visual Sudoku [20] | ✗ | ✗ | ✗ | ✗ | ✗ |
| Handwritten Formula [13] | ✗ | ✗ | ✗ | ✗ | ✗ |
| Smokers & Friends [21] | ✓ | ✗ | ✗ | ✗ | ✗ |
| CLEVR [22] | ✓ | ✓ | ✗ | ✗ | ✓ |
| BlocksWorld [7] | ✓ | ✗ | ✗ | ✓ | ✗ |
| Atari Games [23] | ✗ | − | − | ✓ | ✓ |
| Minigrid & Miniworld [24] | − | − | ✗ | − | ✓ |
| BabyAI [25] | ✓ | − | ✗ | ✓ | ✓ |
| HighWay [26] | ✓ | ✗ | ✓ | ✗ | ✓ |
| **LogiCity (Ours)** | ✓ | ✓ | ✓ | ✓ | ✓ |

distinct train/test agent sets to govern their behaviors in the simulation. To render the environment into diverse RGB images, we leverage foundational generative models [1, 30–32]. Since our modular framework enables seamless configuration adjustments, researchers can explore compositional generalization by changing agent sets while keeping abstractions fixed, or study adaptation to new and more complex abstractions by altering rules and concepts.

To exercise different aspects of NeSy AI, we use LogiCity to design tasks for both sequential decision-making (SDM) and visual reasoning. In the SDM task, algorithms need to navigate a lengthy path (> 40 steps) with minimal trajectory cost, considering rule violations and step-wise action costs. This involves planning amidst complex scenarios and interacting with multiple dynamic agents. For instance, decisions like speeding up may incur immediate costs but could lead to a higher return in achieving the goal sooner. Notably, our SDM task is also unique in that training and testing agent compositions are different, requiring an agent to learn the abstractions and generalize to new compositions. Contrarily, the visual reasoning task focuses on single-step reasoning but features high-dimensional RGB inputs. Algorithms must perform sophisticated abstract reasoning to predict actions for all agents with high-level perceptual noise. Across both tasks, we vary reasoning complexity to evaluate the algorithms' ability to adapt and learn new abstractions.

While we show that existing NeSy approaches [7, 11] perform better in learning abstractions, both from scratch and continually, more complex scenarios from LogiCity still pose significant challenges for prior arts [7, 10, 11, 33–39]. On the one hand, LogiCity raises the abstract reasoning complexity with long-horizon multiple agents scenario, which have not been adequately addressed by current methods. Besides, it also highlights the difficulty of learning complex abstractions from high-dimensional data even for one-step reasoning. On the other hand, LogiCity provides flexible ground-truth symbolic abstractions, allowing for the new methods to be gradually designed, developed, and validated. Therefore, we believe LogiCity represents a crucial step for advancing the next generation of NeSy AI capable of sophisticated abstract reasoning and learning.

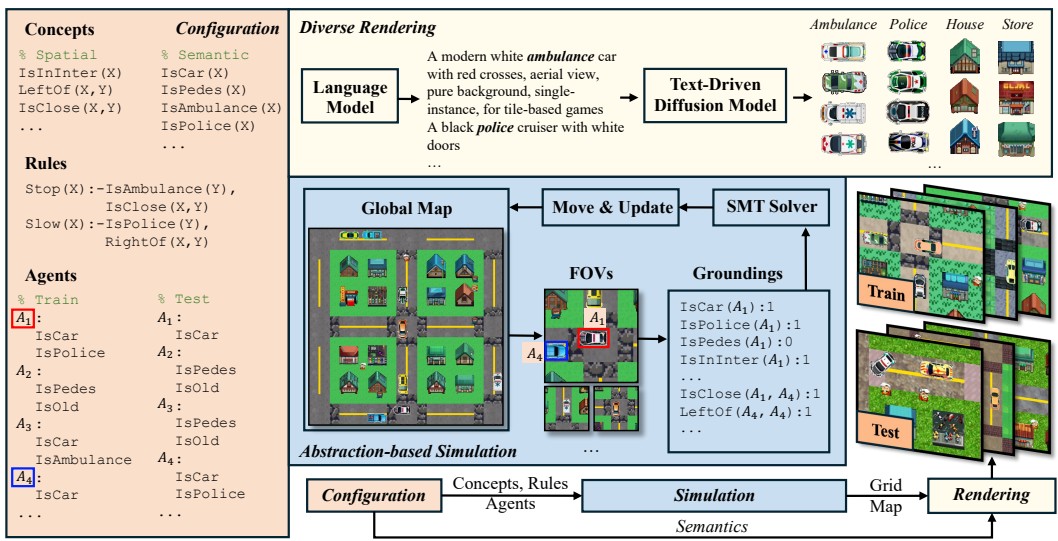

Figure 1: LogiCity employs *abstract* concepts and rules, allowing different agent sets to address compositional generalization. Its modular structure enables users to modify *abstractions* flexibly.

## 2 Related Works

### 2.1 Neuro-Symbolic AI

NeSy AI systems aim to integrate formal logical reasoning into deep neural networks. We distinguish these systems into two categories: *deductive* methods and *inductive* methods.

**Deductive Methods** typically operate under the assumption that the underlying symbolic structure and the chain of deductive reasoning (rules) are either known [8, 9, 21, 40–42] or can be generated by an oracle [17, 18, 43, 44]. Some of these approaches constructed differentiable logical loss functions that constrain the training of deep neural networks [40, 41]. Others, such as DeepProbLog [8], have formulated differential reasoning engines, thus enabling end-to-end learning [45–47]. Recently, Large Language Models (LLMs) have been utilized to generate executable code [17, 43, 44] or planning abstractions [48], facilitating the modular integration of the grounding networks. Despite their success, deductive methods sidestep or necessitate the laborious manually engineered symbolic structure, which potentially limits their applicability in areas lacking formalized knowledge.

**Inductive Methods** focus on generating the symbolic structure either through supervised learning [10, 11, 36, 49–51] or by interacting with the environment [52–54]. One line of research explicitly searches the rule space, such as ∂ILP [49], Difflog [55], and Popper [10]. However, as the rule space can be exponentially large for *abstractions*, these methods often result in prolonged search times. To address this, some strategies incorporate neural networks to accelerate the search process [11, 50, 51]. Another avenue of inductive methods involves designing logic-inspired neural networks where rules are implicitly encoded within the learned weights [7, 20, 56, 57], such as SATNet [20] and Neural Logic Machines (NLM) [7]. While these methods show promise for scalability and generalization, their applications have been predominantly limited to overly simplistic test environments.

### 2.2 Games and Simulations

Various gaming environments [23–25, 27] have been developed to advance AI agents. Atari games [23], for instance, provide diverse challenges ranging from spatial navigation in "Pac-Man" to real-time strategy in "Breakout". More complex games include NetHack [27], StarCraft II [58], and MineCraft [28], where an agent is required to do strategic planning and resource management. LogiCity shares similarities with these games in that agent behavior is governed by rules. Especially, LogiCity can be viewed as a Rogue-like gaming environment [27], where maps and agent settings could be randomly generated in different runs. However, our simulator is uniquely tailored for the

NeSy AI community because: (1) LogiCity provides formal symbolic structure in FOL, enhancing the validation and design of NeSy frameworks. (2) Since FOL predicates and rules are abstractions, a user can arbitrarily customize the composition of the world, introducing adversarial scenarios. (3) LogiCity also supports customizable reasoning complexity through flexible configuration settings. Another key difference between LogiCity and most games [23, 27, 58] is that the behavior of non-player characters (NPCs) in LogiCity is governed by global logical constraints rather than human-engineered behavior trees [59–62]. This design enables NPCs to automatically commit to actions that ensure global rule satisfaction, without the need for manual scripting. Moreover, compared to these games, LogiCity is closer to real urban life, offering a more practical scenario.

Addressing the need for realism, autonomous driving (AD) simulators [26, 63–68] deliver high-quality rendering and accurate traffic simulations but often adhere to fixed rules for limited sets of *concepts*. Among them, the SCENIC language [66–68] is the closest to LogiCity, which uses Linear Temporal Logic to specify AD scenarios. Unlike SCENIC, LogiCity uses *abstractions* in FOL, which allows for the generation of a large number of cities with distinct agent compositions more easily. Besides, LogiCity goes beyond these AD simulators by introducing a broader range of *concepts* and more complex rules, raising the challenge of sophisticated logical reasoning.

## 3  LogiCity Simulator

The overall framework of LogiCity simulator is shown in Figure 1. In the configuration stage, a user is expected to provide *Concepts*, Rules, and Agent set, which are fed into the *abstraction*-based simulator to create a sequence of urban grid maps. These maps are rendered into diverse RGB images via generative models, including a LLM [1] and a text-driven diffusion model [30].

### 3.1  Configuration and Preliminaries

**Concepts**  consist of $K$ background predicates $\mathcal{P} = \{P_i(\cdot)|i = 1, \ldots, K\}$. In LogiCity, we can define both *semantic* and *spatial* predicates. For example, $\texttt{IsAmbulance}(\cdot)$ is an unary semantic predicate and $\texttt{IsClose}(\cdot, \cdot)$ is a binary spatial predicate. These predicates will influence the truth value of four action predicates $\{\texttt{Slow}(\cdot), \texttt{Normal}(\cdot), \texttt{Fast}(\cdot), \texttt{Stop}(\cdot)\}$ according to certain rules.

**Rules**  consist of $M$ FOL clauses, $\mathcal{C} = \{C_m|m = 1, \ldots, M\}$. Following ProLog syntax [29], an FOL clause $C_m$ can be written as:

$$\texttt{Stop(X)} : -\texttt{IsClose(X, Y)} \wedge \texttt{IsAmbulance(Y)} \,,$$

where $\texttt{Stop(X)}$ is the *head*, and the rest after ": −" is the *body*. $\texttt{X}, \texttt{Y}$ are variables, which will be *grounded* into specific entities for rule inference. Note that the clause implicitly declares that the variables in the *head* have a universal quantifier ($\forall$) and the other variables in the *body* have an existential quantifier ($\exists$). We assume only action predicates appear in the *head*, both action and background predicates could appear in the *body*.

The concepts $\mathcal{P}$ and rules $\mathcal{C}$ are *abstractions*, which are not tied to specific entities.

**Agents**  serve as the *entities* in the environment, which is used to *ground* the *abstractions*. We use $\mathcal{A} = \{A_n|n = 1, \ldots, N\}$ to indicate all the $N$ agents in a city. Each agent will initially be annotated with the semantic *concepts* defined in $\mathcal{P}$. For example, an ambulance car $A_1$ is annotated as $A_1 = \{\texttt{IsCar} : \texttt{True}, \texttt{IsAmbulance} : \texttt{True}, \ldots, p\}$, where $p \in \mathbb{R}$ denotes right-of-way priority.

$\mathcal{P}, \mathcal{C}, \mathcal{A}$ make up the configuration of LogiCity simulation. A user can flexibly change any of them seamlessly without modifying the simulation and rendering process.

### 3.2  Simulation and Rendering

As the simulation initialization, a static urban map $\mathbf{M}_s \in \{0, 1\}^{W \times H \times B}$ is constructed, where $W, H$ denotes the width and height. $B$ indicates the number of static semantics in the city, *e.g.*, traffic streets, walking streets, intersections, *etc*. The agents then randomly sample collision-free start and goal locations on the map. These locations are fed into a search-based planner [69] to obtain the global paths that the agents will follow to navigate themselves. On top of the static map, each agent will create an additional dynamic layer, indicating their latest location and planned paths. We use $\mathbf{M}^t \in \{0, 1\}^{W \times H \times (B+N)}$ to denote the full semantic map with all the $N$ agents at time step $t$.

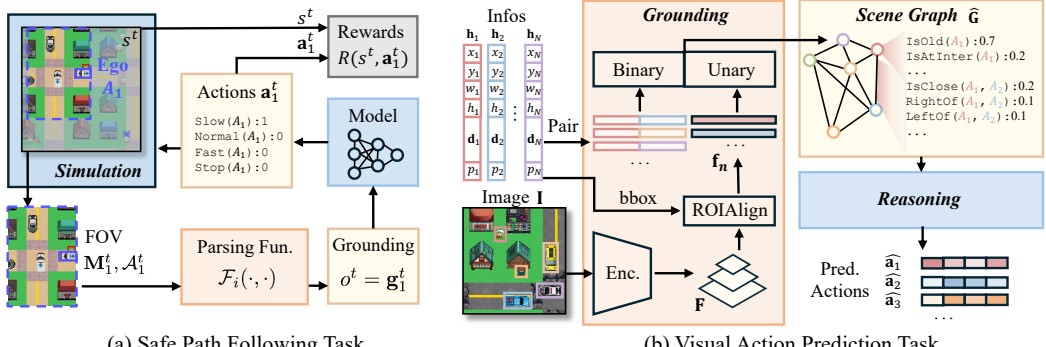

(a) Safe Path Following Task  (b) Visual Action Prediction Task

Figure 2: Demonstration of the Safe Path Following (SPF) and Visual Action Prediction (VAP) tasks in LogiCity. SPF emphasizes sequential decision-making while VAP focuses on one-step sophisticated reasoning on RGB inputs. In the VAP task, we also display the baseline model structure.

During the simulation, each agent $A_n$ only has a limited field-of-view (FOV) of the overall map $\mathbf{M}^t$, which we denote $\mathbf{M}_n^t$. Additionally, FOV agents (self-included) in $\mathbf{M}_n^t$ is obtained and denoted as $\mathcal{A}_n^t$. A group of $K$ pre-defined, binary functions $\{\mathcal{F}_i(\cdot, \cdot)|i = 1, \ldots, K\}$ for the $K$ predicates are then employed to obtain the *grounding* $\mathbf{g}_n^t$ for the ego agent, $\mathbf{g}_n^t = \mathrm{Cat}\big(\mathcal{F}_1(\mathbf{M}_n^t, \mathcal{A}_n^t), \ldots, \mathcal{F}_K(\mathbf{M}_n^t, \mathcal{A}_n^t)\big)$. Here, $\mathrm{Cat}(\cdot)$ denotes concatenation. Assuming we have a total of $N_n$ FOV agents, $\mathbf{g}_n^t$ will be in the shape of $\{0,1\}^{\sum_{i=1}^K N_n^{r_i}}$, where $r_i$ is the *arity* for the $i$-th predicate. Given $\mathbf{g}_n^t$ and the rule clauses $\mathcal{C}$, we leverage an SMT solver [70] to find the truth value of the four *grounded* action predicates, $\mathbf{a}_n^t = \mathrm{SMT}(\mathbf{g}_n^t, \mathcal{C})$. Here, $\mathbf{a}_n^t \in \{0,1\}^4$ denotes the truth value of the *grounded* four action predicates for agent $A_n$ at time $t$. An example of this procedure for $A_1$ is provided in Figure 1. After all the agents take proper actions, we move their location, update the semantic map into $\mathbf{M}^{t+1}$, and repeatedly apply the same procedure. Whenever an agent reaches its goal, the end position becomes the new starting point, a new goal point is randomly sampled, and the navigation is re-started.

To render the binary grid map $\mathbf{M}^t$ into an RGB image $\mathbf{I}^t$ with high visual variance, we leverage foundational generative models [1, 30–32]. We first feed the name of each *semantic concept*, including different types of agents and urban elements, into GPT-4 [1] and asked it to generate diverse descriptions. These descriptions are fed into a diffusion-based generative model [30], which creates diverse icons. These icons will be randomly selected and composed into the grid map landscape to render highly diverse RGB image datasets. Detailed simulation procedure is shown in Appendix C.

## 4 LogiCity Tasks

LogiCity introduced above can exercise different aspects of NeSy AI. For example, as shown in Figure 2, we design two different tasks. The Safe Path Following (SPF) task aims at evaluating sequential decision-making capability while the Visual Action Prediction (VAP) task focuses on reasoning with high-dimensional data. Both tasks assume no direct access to the rule clauses $\mathcal{C}$.

### 4.1 Safe Path Following

SPF requires an algorithm to control an agent in LogiCity, following the global path to the goal while maximizing the trajectory return. The agent is expected to sequentially make a decision on the four actions based on its discrete, partial observations, which should minimize rule violation and action costs. In the following introduction, we assume the first agent, *i.e.*, $A_1$ is the controlled agent.

Specifically, the SPF task can be formulated into a Partially Observable Markov Decision Process (POMDP), which can be defined by the tuples $(S, \mathbb{A}, \Omega, T, Z, R, \gamma)$. The state at time $t$ is the global urban grid, together with all the agents and their conceptual attributes, $s^t = \{\mathbf{M}^t, \mathcal{A}\} \in S$. The action space $\mathbb{A}$ is the 4-dimensional discrete action vector $\mathbf{a}_1^t$. The observation at $t$-th step is the *grounding* of the agent's FOV, $o^t = \mathbf{g}_1^t \in \Omega$, which can be obtained from the parsing functions $\mathcal{F}_i$. State transition $T(s^{t+1}|s^t, \mathbf{a}_1^t)$ is the simulation process introduced in Section 3.2. The observation

function $Z(o^{t+1}|s^{t+1}, \mathbf{a}_1^t)$ is a deterministic cropping function. The reward function $R(s^t, \mathbf{a}_1^t)$ is defined as $R(s^t, \mathbf{a}_1^t) = \sum_m^M w_m^r \psi(s^t, \mathbf{a}_1^t, C_m) + w^a \phi(\mathbf{a}_1^t) + w^{\text{overtime}}(t)$, where $w_m^r$ is the weight of rule violation punishment for the $m-$th clause $C_m$. $\psi(\cdot, \cdot, \cdot)$ evaluates if clause $C_m$ is satisfied for agent $A_1$ given $s^t$ and $\mathbf{a}_1^t$. $\phi(\mathbf{a}_1^t)$ indicates action cost at step $t$ and $w^a$ is a normalization factor. $w^{\text{overtime}}(t)$ gives constant punishment if $t$ is larger than the max horizon. Finally, $\gamma$ is a discount factor set to 0.99. An example of SPF is shown in Figure 2 (a), where $A_1$ is the *Ambulance* car in the purple box. The dashed box denotes its FOV, which will be *grounded* by the parsing functions. A model needs to learn to sequentially output action decisions that maximize trajectory return.

Compared to existing reasoning games [23–25], LogiCity's SPF task presents two unique challenges: (1) Different agent configurations $\mathcal{A}$ in training and testing cause distribution shifts in world transitions ($T$). This requires the model to learn the *abstractions* ($\mathcal{P}, \mathcal{C}$) for compositional generalization. For example, training agents could include *ambulance* plus *pedestrian* and *police* car plus *pedestrian*. In testing, the algorithm may need to plan with *ambulance* plus *police* car. (2) LogiCity supports more realistic multi-agent interaction. For instance, if the controlled agent arrives at an intersection later than other agents, it must wait, resulting in a lower trajectory return; if it speeds up to arrive earlier, others yield, ending up with a higher score. This encourages learning both ego rules and world transitions with multiple agents (how to plan smartly by forecasting).

## 4.2 Visual Action Prediction

Unlike SPF, which is long-horizon and assumes access to the *groundings*, the VAP task is step-wise and requires reasoning on high-dimensional data [13, 20]. As shown in Figure 2 (b), inputs to VAP models include the rendered image $\mathbf{I}$ (We omit the time superscript $t$ here) and information for $N$ agents $[\mathbf{h}_1, \ldots, \mathbf{h}_N] \in \mathbb{R}^{N \times 9}$, where $\mathbf{h}_n = [x_n, y_n, w_n, h_n, \mathbf{d}_n, p]^\top$ consists of location $(x_n, y_n)$, scale $(w_n, h_n)$, one-hot direction $\mathbf{d}_n \in \mathbb{R}^4$, and normalized priority $p$. During training, the models learn to reason and output the action vectors $\hat{\mathbf{a}}_n$ for all the $N$ agents with ground-truth supervision. During test, the models are expected to predict the actions for different sets of agents.

This task is approached as a two-step graph reasoning problem [7, 38]. As illustrated in Figure 2 (b), a grounding module first predicts interpretable *grounded* predicate truth values, which are then used by a reasoning module to deduce action predicates. To be more specific, a visual encoder [2, 71] first extracts global features $\mathbf{F}$ from $\mathbf{I}$. Agent-centric regional features are derived from ROIAlign [72], which resizes the image-space bounding boxes to match the feature scale and then crops the global feature using bilinear interpolation. The resulting regional features for each agent, denoted as $\mathbf{f}_n$, are fed into unary prediction heads to generate unary predicate *groundings*. Meanwhile, binary prediction heads utilize paired agent information to predict binary predicates. Together, the *groundings* form a scene graph $\hat{\mathbf{G}}$, which a graph reasoning engine [7, 38] uses to predict actions $\hat{\mathbf{a}}_n$.

Similar to the SPF task, the VAP task also features different train/test agent compositions, necessitating the model's ability to learn *abstractions*. Additionally, unlike reasoning on structured, symbolic knowledge graphs [7, 11, 21], the diverse visual appearances in LogiCity introduce high-level perceptual noise, adding an extra challenge for reasoning algorithms.

## 5 Experiments

### 5.1 Safe Path Following

We first construct a ground-truth rule-based agent as Oracle and a Random agent as the worst baseline, showing their results in Table 2. Two branches of methods are considered here, behavior cloning (BC) and reinforcement learning (RL), respectively. All the experiments in SPF are conducted on a single NVIDIA RTX 3090 Ti GPU with 32 AMD Ryzen 5950X 16-core processors.

**Baselines.** In the BC branch, we provide oracle trajectories as demonstration and consider the inductive logical programming (ILP) algorithms [10], including symbolic ones [10, 36] and NeSy ones [7, 11]. We also construct a multi-layer perceptron (MLP) and graph neural networks (GNN) [38] as the pure neural baselines. In the RL track, we first build neural agents using various RL algorithms, including on-policy [33, 34], off-policy [7, 35] model-free approaches and model-based algorithms [37, 39]. Since most of the existing NeSy RL methods [52, 53] are carefully engineered for simpler environments, we find it hard to incorporate them into our LogiCity environment. To

Table 2: Empirical results of different methods in SPF task. TSR denotes trajectory success rate (most crucial) and DSR indicates decision success rate. † means Popper timed out. ‡ indicates conflict rules will be inducted for different actions. See our website and Appendix F for episode visualizations.

| Supervision | Mode\Model | Easy | | | Medium | | | Hard | | | Expert | | |
|---|---|---|---|---|---|---|---|---|---|---|---|---|---|
| | Metric | TSR | DSR | Score | TSR | DSR | Score | TSR | DSR | Score | TSR | DSR | Score |
| N/A | Oracle | 1.00 | 1.00 | 8.51 | 1.00 | 1.00 | 8.45 | 1.00 | 1.00 | 9.63 | 1.00 | 1.00 | 4.33 |
| | Random | 0.07 | 0.00 | 0.00 | 0.06 | 0.00 | 0.00 | 0.04 | 0.01 | 0.00 | 0.05 | 0.06 | 0.00 |
| Behavior Cloning | Popper [10] | **1.00** | 1.00 | 8.51 | N/A† | N/A† | N/A† | N/A† | N/A† | N/A† | N/A† | N/A† | N/A† |
| | MaxSynth [36] | **1.00** | 1.00 | 8.51 | 0.25 | 0.67 | 3.18 | 0.15 | 0.60 | 2.96 | 0.09 | 0.21 | 0.37 |
| | HRI [11] | 0.37 | 0.78 | 4.40 | **0.48** | 0.70 | 4.75 | 0.08 | 0.15 | 0.59 | N/A‡ | N/A‡ | N/A‡ |
| | NLM [7] | 0.75 | 1.00 | 7.29 | 0.30 | 0.67 | 3.24 | **0.24** | 0.27 | 2.00 | **0.22** | 0.38 | 0.99 |
| | GNN [38] | 0.26 | 0.39 | 2.58 | 0.17 | 0.24 | 1.31 | 0.19 | 0.39 | 2.19 | 0.19 | 0.32 | 0.84 |
| | MLP | 0.61 | 0.63 | 4.80 | 0.20 | 0.19 | 1.22 | 0.12 | 0.13 | 0.81 | 0.10 | 0.19 | 0.25 |
| Reinforcement Learning | NLM-DQN [7, 35] | **0.53** | 0.96 | 5.93 | **0.47** | 0.67 | 4.40 | **0.29** | 0.40 | 2.69 | **0.15** | 0.35 | 0.62 |
| | MB-shooting [37] | 0.24 | 0.44 | 2.55 | 0.20 | 0.17 | 1.18 | 0.16 | 0.17 | 1.26 | 0.13 | 0.11 | 0.37 |
| | DreamerV2 [39] | 0.07 | 0.43 | 2.86 | 0.02 | 0.21 | 0.67 | 0.00 | 0.30 | 1.45 | 0.12 | 0.06 | 0.41 |
| | DQN [35] | 0.35 | 0.89 | 4.80 | 0.42 | 0.59 | 3.72 | 0.09 | 0.12 | 0.63 | 0.07 | 0.24 | 0.37 |
| | PPO [33] | 0.33 | 0.36 | 2.83 | 0.09 | 0.25 | 0.88 | 0.02 | 0.38 | 1.57 | 0.12 | 0.08 | 0.38 |
| | A2C [34] | 0.10 | 0.16 | 1.00 | 0.06 | 0.29 | 1.07 | 0.00 | 0.14 | 0.46 | 0.12 | 0.09 | 0.34 |

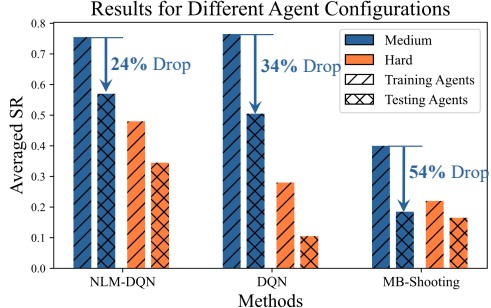

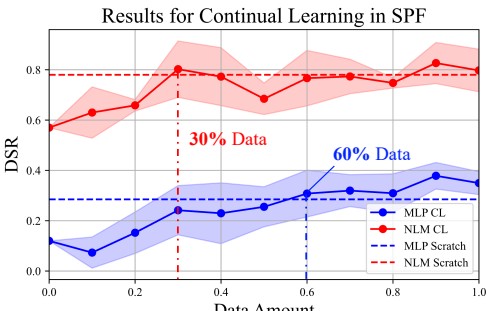

Figure 3: Results for different agent configurations in medium and hard modes of SPF task. We report the average of DSR and TSR here.

Figure 4: Continual learning results for MLP and NLM [7]. The results achieved by training from scratch are reported in dashed lines.

introduce NeSy AI in the RL track, we develop a new Q-learning agent based on NLM [7], which we denote as NLM-DQN [7, 35]. For more details, please see Appendix A.

**Modes and Datasets.** As shown in Table 2, we provide four modes in the SPF task, namely easy, medium, hard, and expert. From easy to medium to hard mode, we progressively introduce more *concepts* and more complex rules, constraining only Stop action. The expert mode constrains all four actions with the most complex rule sets. More details are included in Appendix B.

**Metrics.** We consider three metrics in this task. Trajectory Success Rate (TSR) evaluates if an agent can reach its goal within the allotted time. It is defined as $\text{TSR} = \frac{\sum_i^{T^{\text{all}}} \text{Succ}_i}{T^{\text{all}}}$, where $T^{\text{all}}$ is the total number of episodes, and $\text{Succ}_i = 1$ if the $i$-th episode is completed within twice the oracle steps without rule violations, and $\text{Succ}_i = 0$ otherwise. Decision Success Rate (DSR) assesses if an agent adheres to all rules. It is defined as $\text{DSR} = \frac{\sum_i^{T^{\text{all}}} \text{Dec}_i}{T^{\text{all}}}$, where $\text{Dec}_i = 1$ if the $i$-th episode has at least one rule-constrained step and the agent does not violate any rules throughout, regardless of task completion, and $\text{Dec}_i = 0$ otherwise. The score metric is the averaged trajectory return over all episodes minus the return of a random agent. Among them, TSR is the most crucial.

### 5.1.1 Empirical Evaluation

We present the empirical results in Table 2, showing LogiCity's ability to vary reasoning complexity. In the BC track, symbolic methods [10, 36] perform well in the *easy* mode but struggle with more

Table 3: Empirical results of different methods and settings in VAP task (Modular is more crucial). We report the recall rate for each action, averaged accuracy (aAcc.), and weighted accuracy (wAcc.).

| Supervision | Mode
Num. Actions
Config | Model | Easy
3042
Slow | 3978
Normal | 7220
Stop | -
aAcc. | -
wAcc. | Hard
4155
Slow | 2882
Normal | 715
Fast | 6488
Stop | -
aAcc. | -
wAcc. |
|---|---|---|---|---|---|---|---|---|---|---|---|---|---|
| Modular | Fixed | GNN [38] | 0.45 | 0.63 | 0.54 | 0.54 | **0.53** | 0.44 | 0.47 | 0.09 | 0.57 | 0.49 | 0.23 |
| | | NLM [7] | 0.31 | 0.57 | 0.75 | 0.61 | 0.49 | 0.39 | 0.54 | 0.11 | 0.48 | 0.45 | **0.24** |
| | Random | GNN [38] | 0.52 | 0.63 | 0.43 | 0.51 | 0.54 | 0.26 | 0.51 | 0.19 | 0.63 | 0.48 | 0.28 |
| | | NLM [7] | 0.54 | 0.53 | 0.67 | 0.60 | **0.56** | 0.15 | 0.41 | 0.35 | 0.57 | 0.41 | **0.36** |
| E2E | Fixed | GNN [38] | 0.76 | 0.69 | 0.98 | 0.85 | **0.78** | 0.46 | 0.62 | 0.27 | 0.99 | 0.72 | 0.40 |
| | | NLM [7] | 0.78 | 0.47 | 1.00 | 0.83 | 0.71 | 0.33 | 0.69 | 0.37 | 1.00 | 0.71 | **0.46** |
| | Random | GNN [38] | 0.88 | 0.64 | 1.00 | 0.87 | **0.82** | 0.14 | 0.66 | 0.52 | 1.00 | 0.65 | **0.54** |
| | | NLM [7] | 0.90 | 0.53 | 1.00 | 0.85 | 0.79 | 0.25 | 0.67 | 0.45 | 1.00 | 0.69 | 0.50 |

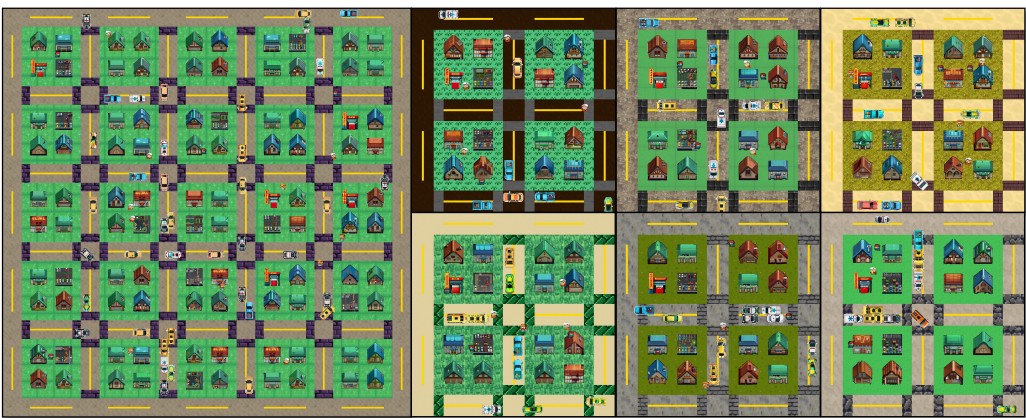

Figure 5: Diverse renderings from LogiCity. Note that every city has distinct agent compositions.

complex scenarios from the *medium* mode. NeSy rule induction methods [7, 11] outperform pure neural MLP/GNN approaches. In the RL track, off-policy methods [7, 35, 37, 39] are more stable and effective than on-policy methods [33, 34] due to the high variance in training episodes affecting policy learning. Additionally, NeSy framework [7, 35] outperform pure neural agents [35, 37] by finding better representations from *abstract* observations. To illustrate the compositional challenge in LogiCity, we compare results across different agent sets in Figure 3. Models trained on the training agent configuration show significant performance drops when transferred to test agents, but NeSy methods [7, 35] are less affected. We discuss more observation spaces in Appendix D.

### 5.1.2 Continual Learning

Using LogiCity, we also examine how much data different models need to continually learn new *abstract* rules. We initialize models with the converged weights from *easy* mode and progressively provide data from *medium* mode rules. The results from three random runs for MLP and NLM [7] are shown in Figure 4, alongside results from models trained from scratch. NLM reaches the best result with 30% of the target domain data, demonstrating superior continual learning capabilities.

### 5.2 Visual Action Prediction

**Baselines.** As there exists very limited literature [73] studying FOL reasoning on RGB images, we self-construct two baselines using GNN [38] and NLM [7] as the reasoning engine, respectively. For fairness, we use the same visual encoder [2, 71] and hyperparameter configurations. We train and test all the models on a single NVIDIA H100 GPU. See Appendix A for more details.

**Settings.** We explore four distinct training settings for the two methods. Regarding supervision signals, modular supervision offers ground truth for both scene graphs and final actions, training the two modules separately. This setting requires interpretable meanings of the scene graph elements,

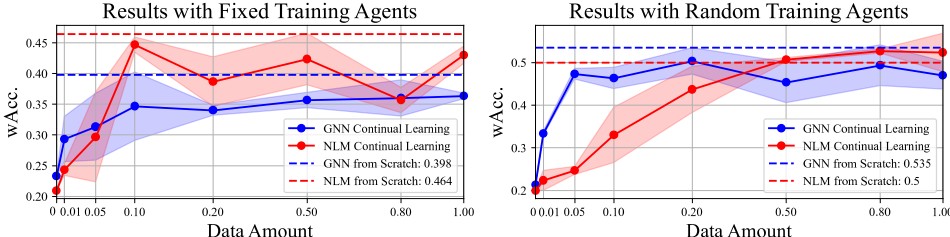

Figure 6: Continual learning results of GNN [38] and NLM [7] in the VAP task. The mean results from three random runs are displayed in solid lines and the variance is reported as the semi-transparent regions. We also show the results of the models trained from scratch using 100% data in dashed lines.

which is crucial. We also explore end-to-end supervision (E2E), which provides guidance only on the final actions. For the training agent sets, we experiment with both fixed and random settings.

**Modes and Datasets.** We present two modes for VAP task, namely *easy* and *hard*. In *easy* mode, rules constrain only `Slow` and `Stop` actions with few *concepts*. The *hard* mode includes the *easy abstractions* and additional constraints for all four actions, with a natural data imbalance making the `Fast` action rarer. We display some examples in Figure 5. More details are included in Appendix B.

**Metrics.** We first report the action-wise recall rate (true positives divided by the number of samples). The average accuracy (aAcc.) is the correct prediction rate across all the test samples. To highlight the data imbalance issue, we also introduce weighted accuracy (wAcc.), defined as wAcc. $= \frac{\sum_a \text{Recall}^a / N^a}{\sum_a \frac{1}{N^a}}$, where $\text{Recall}^a$ is the recall rate for action `a` and $N^a$ is the number of samples for action `a`. This metric assigns larger weights to less frequent actions.

### 5.2.1 Empirical Evaluation

The empirical results for the VAP task are shown in Table 3, highlighting LogiCity's ability to adjust reasoning complexity. We observe that while GNN [38] slightly outperforms NLM [7] in the *easy* mode, NLM excels in the *hard* mode. We also find that random agent configurations improve the performance of both methods. The data imbalance poses an additional challenge, with the `Stop` action having $2\times \sim 6\times$ higher recall than the `Fast` action. Besides, the modular setting proves more challenging than the end-to-end (E2E) setting, as the modular system is more sensitive to perceptual noise. We further investigate this issue quantitatively in Appendix E.

### 5.2.2 Continual Learning

Similar to the SPF task, we investigate how much data the methods need to continually learn abstract rules in the VAP task. The models pre-trained in *easy* mode are used as the initial weights, which are continually trained with different sets of data from the *hard* mode. The data are sampled for 3 times and the mean and variance of the results are reported in Figure 6, where we also report the results from the models trained with 100% data from scratch as dashed lines ("upper bound"). We observe that the two methods could struggle to reach their "upper bound" if fixed training agents are used. For the random agent setting, NLM [7] could progressively learn new rules and reach its "upper bound" with around 50% data while GNN fails even with 100% data.

### 5.2.3 How Do LLMs and Human Perform in LogiCity?

Recent years have witnessed the increasing use of LLMs for decision making [74–77], concept understanding [78–80], and logical reasoning [81–84]. In this section, we investigate the capability of LLMs [1] and Human to solve the (subset of) VAP task in LogiCity through in-context demonstrations. Since we focus only on logical reasoning, true *groundings* are provided in natural language documents without perceptual noise. Specifically, we first convert every scene (frame) into a paragraph of natural language description (see Figure B for examples). For each entity within the frame, given the scene descriptions, we ask LLMs to decide its next action from options ("A. Slow", "B. Normal", "C. Fast", "D. Stop"). Since the entire test set of VAP is huge, we randomly selected a "Mini" test with 117 questions about the concept `IsTiro` and `IsReckless`. To construct demonstrations for

in-context learning, we randomly choose 5-shot samples from the training document used by human participants[2] and provide question-answer pairs. The performance of Human, `gpt-4o` (GPT-4o), `gpt-4o-mini` (GPT-4o mini), `gpt-4-turbo-2024-04-09` (GPT-4), and `gpt-3.5-turbo-1106` (GPT-3.5) on VAP hard mode test sets are reported in Table 4, where the random sampling results for options are also provided for reference. Based on experts' evaluation, we also display the "hardness" of correctly answering each of the choice, where †, ††, and ††† denote "easy", "medium", and "hard".

We observe that the latest GPT-4o shows significantly better in-context learning capability than previous GPT-4 and GPT-3.5, surpassing them by over 20% in terms of overall accuracy. The results also demonstrate the importance of model scale for reasoning task, where GPT-4o mini falls far behind GPT-4o. However, it is still far from the inductive logical reasoning capability of Human, especially for harder reasoning choices like "Stop". Interestingly, the distribution of the decisions demonstrates that GPT-4 has a strong bias towards a conservative decision, which tends to

Table 4: Action prediction accuracy of different LLMs in the VAP task hard mode. †, ††, and ††† denote different logical reasoning hardness.

| Method | Slow†† | Normal†† | Fast† | Stop††† | Overall |
|---|---|---|---|---|---|
| Human | **95.0** | **92.9** | 48.0 | **83.3** | **81.2** |
| GPT-4o | 20.0 | 84.1 | 80.0 | 32.2 | 59.0 |
| GPT-4 | 75.0 | 57.9 | 25.3 | 2.2 | 39.6 |
| GPT-3.5 | 0.0 | 82.5 | 16.0 | 0.0 | 33.0 |
| GPT-4o mini | 0.0 | 2.4 | **86.7** | 40.0 | 29.6 |
| Random | 21.0 | 23.8 | 28.8 | 27.3 | 25.3 |

predict "Slow" action. GPT-4o is better at reasoning in the context, yet they still tend to use common sense knowledge (*e.g.*, Reckless cars always drive fast). In contrast, human participants tend to learn LogiCity's rules through formal verification, where hypotheses are verified and refined based on training documents. Yet, due to the challenging nature of logical induction, human has made mistakes in learning rules of "Stop" (they concluded more general rules than GT), which affects the accuracy of "Fast". This suggests a promising future research direction that could involve coupling LLMs with a formal inductive logical reasoner [10, 36], creating a generation-verification loop. Another intriguing direction is using the LogiCity dataset to conduct Direct Preference Optimization (DPO) [85].

## 6  Discussions

**Conclusion.**   This work presents LogiCity, a new simulator and benchmark for the NeSy AI community, featuring a dynamic urban environment with various *abstractions*. LogiCity allows for flexible configuration on the *concepts* and FOL rules, thus supporting the customization of logical reasoning complexity. Using the LogiCity simulator, we present sequential decision-making and visual reasoning tasks, both emphasizing *abstract* reasoning. The former task is designed for a long-horizon, multi-agent interaction scenario while the latter focuses on reasoning with perceptual noise. With exhaustive experiments on various baselines, we show that NeSy frameworks [7, 11] can learn *abstractions* better, and are thus more capable of the compositional generalization tests. Yet, LogiCity also demonstrates the challenge of learning *abstractions* for all current methods, especially when the reasoning becomes more complex. Specifically, we highlight the crucial difficulty of long-horizon *abstract* reasoning with multiple agents and that *abstract* reasoning with high dimensional data remains hard. On the one hand, LogiCity poses a significant challenge for existing approaches with sophisticated reasoning scenarios. On the other hand, it allows for the gradual development of the next-generation NeSy AI by providing a flexible environment.

**Limitations and Social Impact.**   One limitation of our simulator is the need for users to pre-define rule sets that are conflict-free and do not cause deadlocks. Future work could involve distilling real-world data into configurations for LogiCity, streamlining this definition process. Currently, LogiCity does not support temporal logic [41]; incorporating temporal constraints on agent behaviors is intriguing. The simulation in LogiCity is deterministic, introducing randomness through fuzzy logic deduction engines [8, 9] could be interesting. For the autonomous driving community [26, 63], LogiCity introduces more various *concepts*, requiring a model to plan with *abstractions*, thus addressing a new aspect of real-life challenges. Enhancing the map of LogiCity and expanding the action space could make our simulator a valuable test bed for them. Additionally, since the ontologies and rules in LogiCity can be easily converted into Planning Definite Domain Language (PDDL), LogiCity has potential applications in *multi-agent* task and motion planning [12, 86]. A potential negative social impact is that rules in LogiCity may not accurately reflect our real life, introducing sim-to-real gap.

---

[2]Since human are able to learn from more samples without the context window limitation, they have read more training documents than LLMs for a more comprehensive understanding of LogiCity.

## Acknowledgment

We acknowledge the support of the Air Force Research Laboratory (AFRL), DARPA, under agreement number FA8750-23-2-1015. This work used Bridges-2 at PSC through allocation cis220039p from the Advanced Cyberinfrastructure Coordination Ecosystem: Services & Support (ACCESS) program which is supported by NSF grants #2138259, #2138286, #2138307, #2137603, and #213296. This work was also supported, in part, by Individual Discovery Grants from the Natural Sciences and Engineering Research Council of Canada, and the Canada CIFAR AI Chair Program. We thank the Microsoft Accelerating Foundation Models Research (AFMR) program for providing generous support of Azure credits. We express sincere gratitude to all the human participants for their valuable time and intelligence devotion in the this research. The authors would also like to express sincere gratitude to Jiayuan Mao (MIT), Dr. Patrick Emami (NREL), and Dr. Peter Graf (NREL) for their valuable feedback and suggestions on the early draft of this work.

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

# A   Detailed Baseline Configurations

To make our experiments reproducible, we provided detailed baseline introductions and configurations below. For more details, please refer to our code base.

## A.1   Safe Path Following

In the BC branch, we have considered ILP methods [10, 36], including both symbolic ones [10, 36] and NeSy ones [7, 11]. For them, we convert the demonstration trajectories (step-wise truth value of all the predicates) into *facts* and conduct rule learning. Popper [10] is one of the most performant search-based rule induction algorithm, which uses failure samples to construct hyposithes spaces via answer set programming. It shows better scaling capability than previous template based methods [49]. Since Popper is a greedy approach, it usually costs too much time searching. Maxsynth [36] relax this greedy setting and aims at finding rules in noisy data via anytime solvers. In our experiments, we set 300 seconds (averaged training time for other methods) as the maximum search time for Popper and Maxsynth. For all the other parameters, official default settings are used for fairness. HRI [11] is a hierachical rule induction framework, which utilizes neural tensors to represent predicates and searches the explicit rules by finding paths between predicates. For different modes in LogiCity, we provided the number of background predicates as HRI initialization. All the other parameter settings are kept the same as the original implementation. When constructing the scene graph, we make sure the ratio of positive and negative samples is 1:1. For the other NLM [7] is an implicity rule induction method, which proposed a FOL-inspired network structure. The learnt rules are implicity stored in the network weights. For different modes in LogiCity, we provided the number of background predicates as NLM initialization. Across different modes, we used the same hyperparameters, *i.e.*, the output dimension of each layer is set to $8$, the maximum depth is set to $4$, and the breadth is $3$. For the baselines above, we used their official optimizer during training. In addition, we constructed pure neural baselines, including an MLP and a GNN [38], both having two hidden layers with ReLU activations. In the easy and medium modes, the dimensions of the hidden layers are $128 \times 64$ and $64 \times 64$. In the hard and expert modes, the dimensions of the hidden layers are $128 \times 64$ and $64 \times 128$. These self-constructed baselines are trained with Adam optimizer [87]. For more details, please refer to our open-sourced code library.

In the RL branch, we first build neural agents using different algorithms, which are learnt by interaction with the environment. A2C [34] is a synchronous, deterministic variant of Asynchronous Advantage Actor Critic (A3C) [34], which is an on-policy framework. It leverages multiple workers to replace the use of replay buffer. Proximal Policy Optimization (PPO) combines the idea in A2C and the trust region optimization in TRPO [88]. Different from these policy gradient-based methods, Deep Q network (DQN) [35] is an off-policy value-based approach, which has been one of the state-of-the-arts in Atari Games [23]. For these three baselines, we used a two-layer MLP as the feature extractor, which has the same structure as the MLP baseline in the BC branch. All the other configurations are borrowed from stable-baselines3 [89]. In addition to these model-free agents, we also considered model-based approaches [37, 39]. MB-shooting [37] uses the learnt world model to evaluate the randomly sampled future trajectories. In our experiments, we used an ensemble of 50 MLPs (with the same structure as above) as the dynamics model. The reward prediction is modeled as a regression problem while the state prediction is a classification problem. During inference, we sample a total of 100 random action sequences with a horizon of 10. DreamerV2 [39] is a more advanced model-based method, which introduced discrete distribution in the latent world representation. We find the official implementation for Atari games [39] is hard to work for LogiCity. Therefore, we have tried our best to carefully tune the parameters, which can be found in our code library. Additionally, we built a NeSy agent [7] based on DQN [35], named as NLM-DQN, which we show the detailed structure in Figure A. The observed *groundings* is first reshaped into a list of predicates, which is fed into NLM to obtain the invented (8) new predicates. Since we are learning ego policy (for the first entity), the first axis of the feature is extracted as the truth value grounded to the ego entity. Then, similar to the vanilla DQN, we construct two MLPs to estimate the current Q value and the next Q value, which, together with the current reward, are used to update the model based on Bellman Equation. Despite its simple structure, NLM-DQN has been demonstrated as the most performant baseline in LogiCity SPF task RL branch, showcasing the power of NeSy in terms of complicated abstract reasoning. All the baselines in the RL branch are trained for a total of $200k$ steps in the training environment, the most performant checkpoints in validation environment is

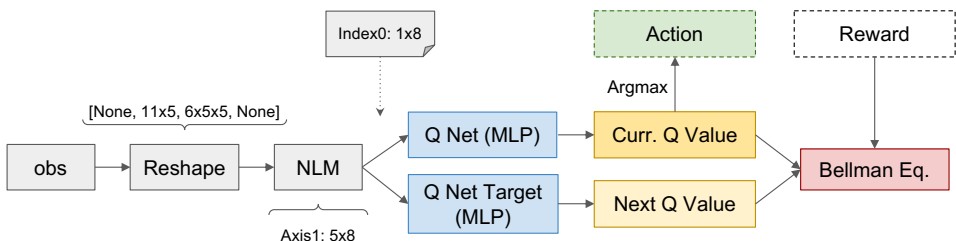

Figure A: Model structure of NLM-DQN [7, 35]. We display the feature dimension for the hard mode for reference.

Table A: All the predicates in LogiCity. We also display which parts of them are involved in each mode of the two tasks.

| Predicates | Arity | Task Description | SPF Easy | SPF Medium | SPF Hard | SPF Expert | VAP Easy | VAP Hard |
|---|---|---|---|---|---|---|---|---|
| IsPedestrian(X) | 1 | Checks if entity X is a pedestrian. | ✓ | ✓ | ✓ | ✓ | ✓ | ✓ |
| IsCar(X) | 1 | Checks if entity X is a car. | ✓ | ✓ | ✓ | ✓ | ✓ | ✓ |
| IsAmbulance(X) | 1 | Checks if entity X is an ambulance. | ✓ | ✓ | ✓ | ✓ | ✓ | ✓ |
| IsBus(X) | 1 | Checks if entity X is a bus. | ✗ | ✓ | ✓ | ✓ | ✓ | ✓ |
| IsPolice(X) | 1 | Checks if entity X is a police vehicle. | ✗ | ✗ | ✓ | ✓ | ✓ | ✓ |
| IsTiro(X) | 1 | Checks if entity X is a tiro. | ✓ | ✓ | ✓ | ✓ | ✓ | ✓ |
| IsReckless(X) | 1 | Checks if entity X is reckless. | ✗ | ✗ | ✓ | ✓ | ✓ | ✓ |
| IsOld(X) | 1 | Checks if entity X is old. | ✓ | ✓ | ✓ | ✓ | ✓ | ✓ |
| IsYoung(X) | 1 | Checks if entity X is young. | ✗ | ✗ | ✓ | ✓ | ✓ | ✓ |
| IsAtInter(X) | 1 | Checks if entity X is at the intersection. | ✓ | ✓ | ✓ | ✓ | ✓ | ✓ |
| IsInInter(X) | 1 | Checks if entity X is in the intersection. | ✓ | ✓ | ✓ | ✓ | ✓ | ✓ |
| IsClose(X, Y) | 2 | Checks if entity X is close to entity Y. | ✗ | ✗ | ✓ | ✓ | ✓ | ✓ |
| HigherPri(X, Y) | 2 | Checks if entity X has higher priority than entity Y. | ✓ | ✓ | ✓ | ✓ | ✓ | ✓ |
| CollidingClose(X, Y) | 2 | Checks if entity X is about to collide with entity Y. | ✓ | ✓ | ✓ | ✓ | ✓ | ✓ |
| LeftOf(X, Y) | 2 | Checks if entity X is left of entity Y. | ✗ | ✗ | ✓ | ✓ | ✓ | ✓ |
| RightOf(X, Y) | 2 | Checks if entity X is right of entity Y. | ✗ | ✓ | ✓ | ✓ | ✓ | ✓ |
| NextTo(X, Y) | 2 | Checks if entity X is next to entity Y. | ✗ | ✓ | ✓ | ✓ | ✓ | ✓ |

utilized for testing. Note that this is different from existing gaming environments [23, 28, 58], where train/val/test environments have very limited distribution shift.

## A.2 Visual Action Prediction

In the VAP task, we built two baseline models with similar structure, namely GNN [38] and NLM [7]. Across the two models, we used the same grounding framework. Specifically, ResNet50 [2] plus Feature Pyramid Network (FPN) [71] pre-trained on ImageNet [90] is leveraged as the feature encoder. After ROIAlign [72], the resulting regional features are in the shape of $\mathbb{R}^{512}$. The unary predicate heads are three-layer MLPs with BatchNorm1D, ReLU, and Dropout functions. Note that the unary predicates are all about the regional feature, requiring no additional information $\mathbf{h}$. On the other hand, the binary predicates are all about the additional information. We first concatinate the information $\mathbf{h}$ for each pair of entities and used two-layer MLPs to predicate the truth values of binary predicates. For details about the structure of the MLPs, please see our code library. The truth values of unary and binary predicates form a scene graph for the reasoning networks [7, 38] to predict actions. For GNN [38], we used a hidden layer in the dimension of 128. For NLM [7], we employed official implementation, where each logic layer invents 8 new attributes, the maximum depth is set to 4 and the breadth is set to 3. In the end-to-end setting, both methods are trained using AdamW [91]. In the modular setting, the grounding module is trained using Adam [87] while the reasoning module is optimized using AdamW [91]. For all the experiments, we train the models for 30 epochs and test the best performing checkpoint in the validation set. Note that these settings are the same in the two modes of VAP task.

## B  Detailed Task Configurations

The full list of predicates and rules and their descriptions are displayed in Table A and Table B, respectively. Across different modes in the two tasks, the involved predicates and rule clauses are the subsets of these full lists. We introduce the detailed configurations below.

Table B: All the rule clauses and their descriptions in the expert mode of the SPF tasks. The clauses in other modes are the subsets of this full list.

| Rule | Description |
| --- | --- |
| `Stop(X):- Not(IsAmbulance(X)), Not(IsOld(X)), IsAtInter(X), IsInInter(Y).` | If `X` is not an ambulance and not old, and `X` is at an intersection, and `Y` is in an intersection, then `X` should stop. |
| `Stop(X):- Not(IsAmbulance(X)), Not(IsOld(X)), IsAtInter(X), IsAtInter(Y), HigherPri(Y, X).` | If `X` is not an ambulance and not old, and `X` is at an intersection, and `Y` is at an intersection, and `Y` has higher priority than `X`, then `X` should stop. |
| `Stop(X):- Not(IsAmbulance(X)), Not(IsOld(X)), IsInInter(X), IsInInter(Y), IsAmbulance(Y).` | If `X` is not an ambulance and not old, and `X` is in an intersection, and `Y` is in an intersection, and `Y` is an ambulance, then `X` should stop. |
| `Stop(X):- Not(IsAmbulance(X)), Not(IsPolice(X)), IsCar(X), Not(IsInInter(X)), Not(IsAtInter(X)), LeftOf(Y, X), IsClose(Y, X), IsPolice(Y).` | If `X` is not an ambulance and not police, and `X` is a car, and `X` is not in or at an intersection, and `Y` is left of and close to `X`, and `Y` is police, then `X` should stop. |
| `Stop(X):- IsBus(X), Not(IsInInter(X)), Not(IsAtInter(X)), RightOf(Y, X), NextTo(Y, X), IsPedestrian(Y).` | If `X` is a bus, and `X` is not in or at an intersection, and `Y` is right of and next to `X`, and `Y` is a pedestrian, then `X` should stop. |
| `Stop(X):- IsAmbulance(X), RightOf(Y, X), IsOld(Y).` | If `X` is an ambulance, and `Y` is right of `X`, and `Y` is old, then `X` should stop. |
| `Stop(X):- Not(IsAmbulance(X)), Not(IsOld(X)), CollidingClose(X, Y).` | If `X` is not an ambulance and not old, and `X` is close to colliding with `Y`, then `X` should stop. |
| `Slow(X):- Not(Stop(X)), IsTiro(X), IsPedestrian(Y), IsClose(X, Y).` | If `X` should not stop, and `X` is a tiro, and `Y` is a pedestrian, and `X` is close to `Y`, then `X` should slow. |
| `Slow(X):- Not(Stop(X)), IsTiro(X), IsInInter(X), IsAtInter(Y).` | If `X` should not stop, and `X` is a tiro, and `X` is in an intersection, and `Y` is at an intersection, then `X` should slow. |
| `Slow(X):- Not(Stop(X)), IsPolice(X), IsYoung(Y), IsYoung(Z), NextTo(Y, Z).` | If `X` should not stop, and `X` is police, and `Y` is young, and `Z` is young, and `Y` is next to `Z`, then `X` should slow. |
| `Fast(X):- Not(Stop(X)), Not(Slow(X)), IsReckless(X), IsAtInter(Y).` | If `X` should not stop, and `X` should not slow, and `X` is reckless, and `Y` is at an intersection, then `X` should go fast. |
| `Fast(X):- Not(Stop(X)), Not(Slow(X)), IsBus(X).` | If `X` should not stop, and `X` should not slow, and `X` is a bus, then `X` should go fast. |
| `Fast(X):- Not(Stop(X)), Not(Slow(X)), IsPolice(X), IsReckless(Y).` | If `X` should not stop, and `X` should not slow, and `X` is police, and `Y` is reckless, then `X` should go fast. |

## B.1 Safe Path Following

**Modes and Dataset:** Across all modes, we fix the (maximum) number of FOV agents into 5, *i.e.*, $\tilde{N}_1 = 5$. If the number of observed agents are fewer than 5, zero-padding (closed-world assumption) is utilized, otherwise, we neglect the extra agents. The predicates involved in each mode are displayed in Table A. Easy mode includes 7 unary and 2 binary predicates, resulting in an $\sum_i^K \tilde{N}_1^{r_i} = 85$ dimensional grounding vector. Rules involve only *spatial* concepts and constrain the `Stop` action. Medium mode features 8 unary predicates and 4 binary predicates, creating a $\sum_i^K \tilde{N}_1^{r_i} = 140$ dimensional grounding vector. The medium rule sets is extended from the easy mode and incorporate both *spatial* and *semantic* concepts, constraining the `Stop` action. Hard mode contains 11 unary predicates and 6 binary predicates, yielding a $\sum_i^K \tilde{N}_1^{r_i} = 205$ dimensional grounding vector. Rules cover all *spatial* and *semantic* concepts and constrain the `Stop` action. The expert mode constrains all four actions with the most complex rule sets. We provide standard training/validation/test agent configurations and validation/test episodes for all the modes. The training agents cover all the necessary concepts in the rules, while validation and test agents are different and more complex, see our code library for the detailed agent configuration. For each mode, we collect 40 validation episodes and 100 test episodes using corresponding agent distribution, making sure the episodes cover

all the *concepts* and actions. When training the BC branch algorithms, we collected 100 trajectories from the oracle as the demonstration.

**Reward:** During test, the rule violation weight $w^r$ is set to $-10$ for easy, medium, and hard mode across all the $M$ clauses. For expert mode, we set this constant punishment to $-5$. In terms of step-wise action costs $\phi(\mathbf{a}_1^t)$, the easy, medium, and hard modes are configured as follows: `Slow` : $-2$, `Normal` : $0$, `Fast` : $-2$, `Stop` : $-5$. In the expert mode, the costs are `Slow` : $-2$, `Normal` : $-1$, `Fast` : $-2$, `Stop` : $-3$. Note that the action costs will be normalized by the length of the global path. Overtime punishment is set to $-3$ for all the modes. During training, we find that different methods requires different reward functions to work effectively. Therefore, we first fix the action costs and have tried our best to tune the rule violation and overtime punishment for each method. For fairness, NLM-DQN and DQN used the same training reward. For more details about the reward, please see our code library.

### B.2 Visual Action Predication

**Modes:** As shown in Table A, the predicates in the two modes involve the full list. As for the clauses, easy mode only constrains `Stop` and `Slow` actions, setting `Normal` as the default action. Hard mode constrains all the three actions with `Normal` set as the default actions. Note that the rule clauses in hard model is a superset of that in the easy mode.

**Datasets:** In the random agent setting, we randomly generated 100 and 20 cities with different agent compositions for training and validation, respectively. For each city, we run the simulation for 100 steps and only used the data after 10 steps. In the fixed agent setting, we first pre-define different training/validation/test agent compositions. Then, we randomly initialize the cities for 100 times. For each initialization, we run the simulation for 100 steps and only used the data after 10 steps. This process results in training sets with $8.9k$ images (with more than $89k$ entity samples). The models trained with different setting are tested in the same fixed agent setting test sets. See our code library and dataset for detailed agent compositions.

## C Full Procedure of LogiCity Simulation

We provide more details for the simulator here.

**Static Urban Semantics:** There are a total of $B = 8$ static semantics of the urban map, namely "Walking Street", "Traffic Street", "Crossing", "House", "Office", "Garage", "Store", "Gas Station". They are (currently) only used during initialization. Specifically, different types of agents will sample start and goal locations around different static semantics. Pedestrians will move from "House", "Office", and "Store" to "House", "Office", and "Store", while Cars are navigating between "Garage", "Gas Station", and "Store" to "Garage", "Gas Station", and "Store". Besides, the agents use different search algorithms based on these semantics to construct their global paths. Specifically, the pedestrians leverage A* search on the "movable region" of the map $\mathbf{M}_s$, which is defined as the union of Walking Streets and Crossings. In contrast, for cars, since they should move only on the right side of the road in real-world, we first construct the "one-way" road map of the Traffic Streets, which is a directed cyclic graph. Then, we connect the start and goal points to this road map and add them to the graph nodes. Finally, Dijkstra search is employed to construct the shortest path from start node to the goal node, which is the global path for a car.

**Rendering Details:** As introduced above, there exist 8 static semantics. As shown in Table A, LogiCity also involves 9 semantic *concepts* of the agents. Therefore, for each of the 17 semantics, we ask GPT4 [1] to generate 40 diverse descriptions. Then we leverage Stable-Diffusion XL model [30] to generate $\sim 2000$ diverse icons from these descriptions. Finally, we employed a human expert to select $50 \sim 200$ icons for each semantic. For mode details, please see our code library.

In addition, we also present the full procedure of the scene simulation by LogiCity in Algorithm 1.

# D State Space Comparison

In the SPF task, we by default provide the predicate groundings as the observation of the agents, which is *abstract* and could be *lossy* [12]. Thus, we have also tried to provide exact states to the agents in this section. Specifically, we annotate each pixel of the FOV map with the agent semantics and convert the pixels into 2D semantic point clouds. Since these point clouds contain all the information needed for an optimal policy, it serves as the "Exact State" for the ego agent. The results comparison of using abstract (Abs.) and exact (Exa.) states is shown in Table C, where we find using "Exact State" could

Table C: Comparison of results from different state space in the LogiCity SPF task. By default, the observation state is the *groundings* of the predicates, which is abstract (Abs.) and lossy. We also tried to provide exact state (Exa.) as observation, which is the semantic point cloud in the ego agent FOV.

| Mode | | Easy | | | Hard | | |
|---|---|---|---|---|---|---|---|
| Method | Obs | TSR | DSR | Score | TSR | DSR | Score |
| DQN | Abs. | **0.35** | 0.89 | 4.8 | **0.09** | 0.12 | 0.63 |
| | Exa. | 0.12 | 0.329 | 2.1 | 0.01 | 0.56 | 2.69 |
| MB-Shooting | Abs. | **0.24** | 0.44 | 2.55 | **0.16** | 0.17 | 1.26 |
| | Exa. | 0.23 | 0.264 | 2.12 | 0.02 | 0.24 | 1.32 |

be much harder for the agents to learn the abstractions in *easy* mode. In *hard* mode, the agents can easily converge to overly careful policies and fail to complete the task in time. One possible future solution for "Exact State" is to combine bi-level planning [12] with reinforcement learning.

# E Quantitative Perceptual Noise

Compared with structured knowledge graphs [21, 92, 93], the VAP task of LogiCity introduces diverse RGB images, which require models to conduct abstract reasoning with high-level perceptual noise. We quantitatively display the perception accuracy of different *concepts* from the NLM model [7] in Table D. Even with supervision, the averaged recall rate for the concepts is not satisfiable (Note that the errors will actually accumulate, which will be much more worse than the 55% averaged result). Compared with binary predicates, unary predicates need operation on the RGB image, which is thus harder. We also observe that the results are highly-imbalanced across concepts. For example, pedestrains and cars are easy to recognize, but a police/tiro car is extremely hard to be distinguished from normal ones. In terms of binary predicates, `CollidingClose` is the hardest to learn, since it needs to consider all the locations, sizes, and directions of the two entities, while the others only involves positions or priorities. One potential solution to the perceptual noise is borrowing off-the-shelf foundation models [94, 95] for the grounding task.

# F Visualizations

**SPF:** Visualizations of the SPF task episodes are displayed in Figure C. Compared with the training city shown on the left, test cities have different agent compositions. For example, training city only has 2 old man while test cities has 4 such entities, featuring compositional generalization challenge. Compared with pure neural networks, NeSy method (NLM-DQN) can better generalize to unseen compositions. For example, in Episode 92, Step 84, the ego agent sees two `pedestrians InIntersection` with an `Ambulance AtIntersection`, which is an unseen composition during training. DQN fails here, outputting `Normal` action while NLM-DQN succeeds with the correct `Stop` decision. SPF task also features realistic multi-agent interaction. As shown in Episode 93, Step 125, since the two algorithms made different decisions in previous steps, the city will be very different as the other agents are largely affected by the ego actions.

**VAP:** Visualizations of the VAP task examples are shown in Figure D. Compared with GNN [38], the NeSy method NLM [7] can better understand the *abstractions* of LogiCity. For example, `Reckless` cars drives `Normally` when it is `InIntersection`, while other cars should drive `Slow`. We find that GNN [38] shows limitation in understanding such *concept* and rules, making wrong predictions.

---
**Algorithm 1:** LogiCity Simulation

---
**Input:** Concepts $\mathcal{P}$, Rules $\mathcal{C}$, Agents $\mathcal{A}$, Static urban map $\mathbf{M}_s$, Generative models for rendering

Generate $\mathbf{M}_s$ with dimensions $(W, H, B)$ and sample collision-free coordinates for agents
Compute global paths for each agent using search-based planner

**for** *each time step $t$* **do**

    Update $\mathbf{M}^t$ with current agent locations and paths

    **for** *each agent $A_n$* **do**

        Obtain $\mathbf{M}_n^t$ and $\mathcal{A}_n^t$ using cropping function        ▷ Local FOV observation

        Compute $\mathbf{g}_n^t$ by applying $\mathcal{F}_i$ to $\mathbf{M}_n^t$ and $\mathcal{A}_n^t$     ▷ Grounding predicates

        Compute action predicates $\mathbf{a}_n^t$ using SMT solver with $\mathbf{g}_n^t$ and $\mathcal{C}$   ▷ Rule inference

        Move agent based on $\mathbf{a}_n^t$

    Update $\mathbf{M}^{t+1}$             ▷ Update semantic map

    **if** *agent reaches goal* **then**

        Set new goal location and compute new path   ▷ Re-sample goal and re-plan path

Generate concept descriptions using GPT-4 and generate icons using diffusion model
Compose icons into $\mathbf{M}^t$ to create RGB image $\mathbf{I}^t$          ▷ Rendering

**Output:** RGB images of urban grid maps $\mathbf{I}^t$

---

Table D: Quantitative results for concept recognition in the VAP task of LogiCity. We report the recall rate of NLM [7] model for each predicate (with threshold $0.5$ on the predicted probability). The results are obtained from hard mode with random training agents.

| Arity | Unary | | | | | | | | | | | |
|---|---|---|---|---|---|---|---|---|---|---|---|---|
| Predicates | IsPed. | IsCar | IsAmbu. | IsBus | IsPolice | IsTiro | IsReckl. | IsOld | IsYoung | IsAtInter | IsInInter | Avg. |
| Num. Samples | 10680 | 14240 | 1780 | 1780 | 3560 | 1780 | 3560 | 3560 | 5340 | 7490 | 3627 | |
| Recall@0.5 | 0.774 | 0.981 | 0.251 | 0.4 | 0.073 | 0.024 | 0.158 | 0.328 | 0.563 | 0.278 | 0.332 | 0.553 |

| Arity | Binary | | | | | | |
|---|---|---|---|---|---|---|---|
| Predicates | IsClose | HigherPri | CollidingClose | LeftOf | RightOf | NextTo | Avg. |
| Num. Samples | 23660 | 28902 | 500 | 33046 | 28064 | 15495 | |
| Recall@0.5 | 0.783 | 1 | 0.05 | 0.857 | 0.921 | 0.874 | 0.887 |

## In-Context Demonstrations and Example Questions from LogiCity

You are an expert in First-Order-Logic (FOL) Rule induction, the following question-answers are FOL reasoning
↪ examples. Here are 5 demonstrations:

Question: "In the scene you see a total of 12 entities, they are named as follows: Entity_0, Entity_1,
↪ Entity_2, Entity_3, Entity_4, Entity_5, Entity_6, Entity_7, Entity_8, Entity_9, Entity_10, Entity_11.
↪ There exist the following predicates as their attributes and relations: IsPedestrian (arity: 1), IsCar
↪ (arity: 1), IsAmbulance (arity: 1), IsBus (arity: 1), IsPolice (arity: 1), IsTiro (arity: 1), IsReckless
↪ (arity: 1), IsOld (arity: 1), IsYoung (arity: 1), IsAtInter (arity: 1), IsInInter (arity: 1), IsClose
↪ (arity: 2), HigherPri (arity: 2), CollidingClose (arity: 2), LeftOf (arity: 2), RightOf (arity: 2), NextTo
↪ (arity: 2), Sees (arity: 2). The truth value of these predicates grounded to the entities are as follows
↪ (Only the ones that are True are provided, assume the rest are False): IsPedestrian(Entity_1),
↪ IsPedestrian(Entity_2), IsPedestrian(Entity_3), IsPedestrian(Entity_4), IsPedestrian(Entity_5),
↪ IsCar(Entity_0), IsCar(Entity_6), IsCar(Entity_7), IsCar(Entity_8), IsCar(Entity_9), IsCar(Entity_10),
↪ IsCar(Entity_11), IsAmbulance(Entity_0), IsAmbulance(Entity_11), IsPolice(Entity_6), IsPolice(Entity_10),
↪ IsTiro(Entity_9), IsReckless(Entity_8), IsOld(Entity_3), IsOld(Entity_5), IsYoung(Entity_1),
↪ IsYoung(Entity_2), IsAtInter(Entity_5), IsAtInter(Entity_8), IsAtInter(Entity_11), IsInInter(Entity_0),
↪ IsInInter(Entity_6), IsInInter(Entity_10), IsClose(Entity_1, Entity_3), IsClose(Entity_3, Entity_1),
↪ IsClose(Entity_3, Entity_7), IsClose(Entity_4, Entity_10), IsClose(Entity_7, Entity_3), IsClose(Entity_7,
↪ Entity_10), IsClose(Entity_10, Entity_4), IsClose(Entity_10, Entity_7), IsClose(Entity_10, Entity_11),
↪ IsClose(Entity_11, Entity_10), ..., Sees(Entity_1, Entity_3), Sees(Entity_1, Entity_7), Sees(Entity_1,
↪ Entity_10), Sees(Entity_3, Entity_1), Sees(Entity_3, Entity_7), Sees(Entity_3, Entity_10), Sees(Entity_4,
↪ Entity_10), Sees(Entity_4, Entity_11), Sees(Entity_5, Entity_8), Sees(Entity_7, Entity_10), Sees(Entity_7,
↪ Entity_11), Sees(Entity_8, Entity_5), Sees(Entity_10, Entity_1), Sees(Entity_10, Entity_3), Sees(Entity_10,
↪ Entity_7), Sees(Entity_11, Entity_4), Sees(Entity_11, Entity_7), Sees(Entity_11, Entity_10). What is the
↪ next action of entity Entity_9?"
Option: (A) Slow (B) Normal (C) Fast (D) Stop
Answer: B

... (4 more demos not displayed)

Now try your best to first identify the FOL rules from the examples above and then answer the following
↪ question. Your answer should strictly end with the format of single letter: 'Answer: _.'

Question: "In the scene you see a total of 14 entities, they are named as follows: Entity_0, Entity_1,
↪ Entity_2, Entity_3, Entity_4, Entity_5, Entity_6, Entity_7, Entity_8, Entity_9, Entity_10, Entity_11,
↪ Entity_12, Entity_13. There exist the following predicates as their attributes and relations: IsPedestrian
↪ (arity: 1), IsCar (arity: 1), IsAmbulance (arity: 1), IsBus (arity: 1), IsPolice (arity: 1), IsTiro (arity:
↪ 1), IsReckless (arity: 1), IsOld (arity: 1), IsYoung (arity: 1), IsAtInter (arity: 1), IsInInter (arity:
↪ 1), IsClose (arity: 2), HigherPri (arity: 2), CollidingClose (arity: 2), LeftOf (arity: 2), RightOf (arity:
↪ 2), NextTo (arity: 2), Sees (arity: 2). The truth value of these predicates grounded to the entities are
↪ as follows (Only the ones that are True are provided, assume the rest are False): IsPedestrian(Entity_1),
↪ IsPedestrian(Entity_2), IsPedestrian(Entity_3), IsPedestrian(Entity_4), IsPedestrian(Entity_5),
↪ IsPedestrian(Entity_6), IsCar(Entity_0), IsCar(Entity_7), IsCar(Entity_8), IsCar(Entity_9),
↪ IsCar(Entity_10), IsCar(Entity_11), IsCar(Entity_12), IsCar(Entity_13), IsAmbulance(Entity_12),
↪ IsBus(Entity_10), IsPolice(Entity_9), IsPolice(Entity_11), IsTiro(Entity_8), IsReckless(Entity_0),
↪ IsReckless(Entity_7), IsOld(Entity_3), IsOld(Entity_5), IsYoung(Entity_1), IsYoung(Entity_2),
↪ IsYoung(Entity_4), IsAtInter(Entity_8), IsAtInter(Entity_13), IsInInter(Entity_6), IsInInter(Entity_11),
↪ IsClose(Entity_0, Entity_5), IsClose(Entity_1, Entity_3), IsClose(Entity_2, Entity_8), IsClose(Entity_3,
↪ Entity_1), IsClose(Entity_5, Entity_0), IsClose(Entity_5, Entity_6), IsClose(Entity_5, Entity_10),
↪ IsClose(Entity_6, Entity_5), IsClose(Entity_6, Entity_8), IsClose(Entity_6, Entity_13), IsClose(Entity_8,
↪ Entity_2), IsClose(Entity_8, Entity_6), IsClose(Entity_8, Entity_10), IsClose(Entity_10, Entity_5),
↪ IsClose(Entity_10, Entity_8), IsClose(Entity_10, Entity_13), IsClose(Entity_13, Entity_6),
↪ IsClose(Entity_13, Entity_10), HigherPri(Entity_0, Entity_8), HigherPri(Entity_0, Entity_10),
↪ HigherPri(Entity_0, Entity_12), HigherPri(Entity_0, Entity_13), HigherPri(Entity_2, Entity_0),
↪ HigherPri(Entity_2, Entity_8), HigherPri(Entity_2, Entity_10), HigherPri(Entity_2, Entity_13),
↪ HigherPri(Entity_5, Entity_0), HigherPri(Entity_5, Entity_8), HigherPri(Entity_5, Entity_10),
↪ HigherPri(Entity_5, Entity_12), HigherPri(Entity_5, Entity_13), HigherPri(Entity_6, Entity_0),
↪ HigherPri(Entity_6, Entity_8), HigherPri(Entity_6, Entity_10), HigherPri(Entity_6, Entity_12),
↪ HigherPri(Entity_6, Entity_13), HigherPri(Entity_7, Entity_9), HigherPri(Entity_8, Entity_10),
↪ HigherPri(Entity_8, Entity_12), HigherPri(Entity_8, Entity_13), HigherPri(Entity_10, Entity_12),
↪ HigherPri(Entity_10, Entity_13), HigherPri(Entity_12, Entity_13), CollidingClose(Entity_0, Entity_10),
↪ CollidingClose(Entity_7, Entity_9), CollidingClose(Entity_12, Entity_13), LeftOf(Entity_0, Entity_2),
↪ LeftOf(Entity_0, Entity_6), LeftOf(Entity_0, Entity_8), LeftOf(Entity_2, Entity_5), LeftOf(Entity_2,
↪ Entity_13), LeftOf(Entity_3, Entity_1), LeftOf(Entity_5, Entity_8), LeftOf(Entity_6, Entity_2),
↪ LeftOf(Entity_6, Entity_8), LeftOf(Entity_8, Entity_5), LeftOf(Entity_8, Entity_12), LeftOf(Entity_8,
↪ Entity_13), LeftOf(Entity_10, Entity_2), LeftOf(Entity_10, Entity_6), LeftOf(Entity_10, Entity_8),
↪ LeftOf(Entity_12, Entity_0), ..., Sees(Entity_0, Entity_5), Sees(Entity_0, Entity_6), Sees(Entity_0,
↪ Entity_8), Sees(Entity_0, Entity_10), Sees(Entity_0, Entity_12), Sees(Entity_0, Entity_13), Sees(Entity_1,
↪ Entity_3), Sees(Entity_3, Entity_1), Sees(Entity_5, Entity_4), Sees(Entity_6, Entity_0), Sees(Entity_6,
↪ Entity_10), Sees(Entity_6, Entity_12), Sees(Entity_6, Entity_13), Sees(Entity_7, Entity_9), Sees(Entity_8,
↪ Entity_0), Sees(Entity_8, Entity_2), Sees(Entity_8, Entity_6), Sees(Entity_8, Entity_10), Sees(Entity_8,
↪ Entity_13), Sees(Entity_10, Entity_5), Sees(Entity_10, Entity_6), Sees(Entity_10, Entity_8),
↪ Sees(Entity_10, Entity_12), Sees(Entity_10, Entity_13), Sees(Entity_12, Entity_0), Sees(Entity_12,
↪ Entity_6), Sees(Entity_12, Entity_10), Sees(Entity_12, Entity_13), Sees(Entity_13, Entity_0),
↪ Sees(Entity_13, Entity_2), Sees(Entity_13, Entity_5), Sees(Entity_13, Entity_6), Sees(Entity_13, Entity_8),
↪ Sees(Entity_13, Entity_10). What is the next action of entity Entity_11?"
Option: (A) Slow (B) Normal (C) Fast (D) Stop

Figure B: In-context demos and questions for LLM testing. Note that the demos cover all the options and the test questions have different entity compositions. We only display one demo and part of the groundings due to the space limit.

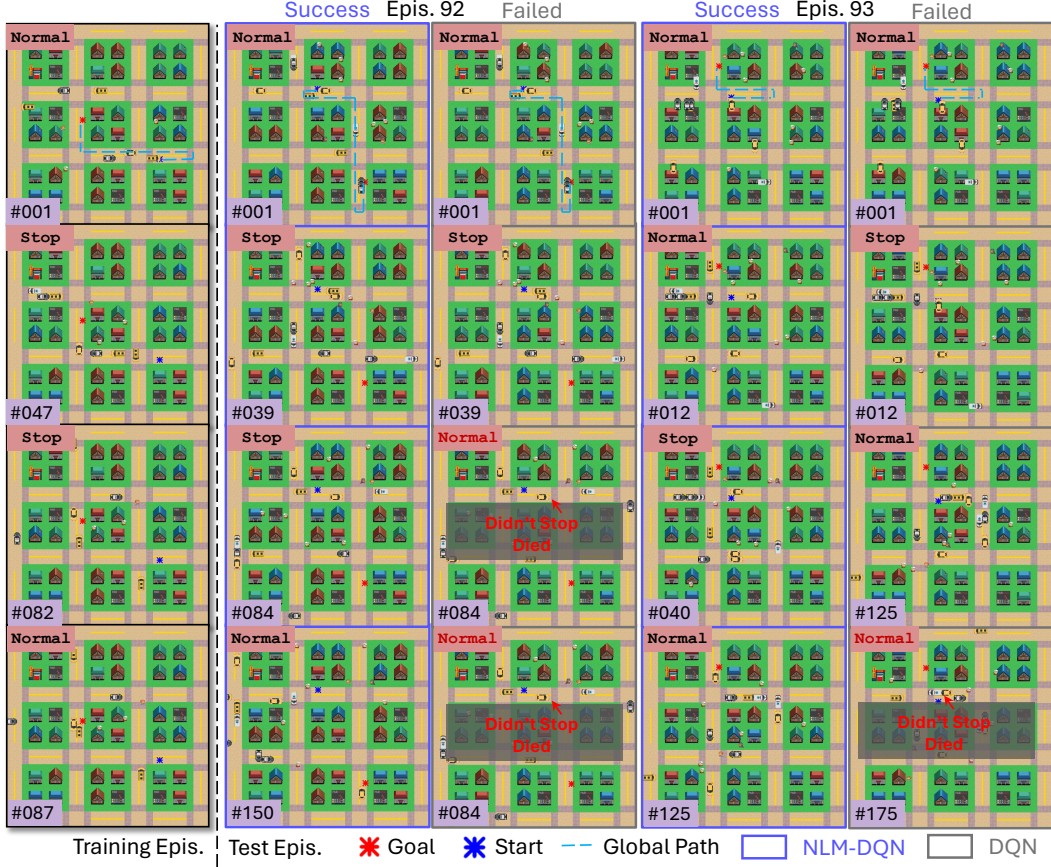

Figure C: Qualitative comparison between NLM-DQN [7, 35] and DQN [35] agents in the hard mode of SPF task. We display the training episode on the left, which has different agent sets from the test, featuring compositional generalization challenge. Compared with the pure neural network, NeSy method [7, 35] is better at abstract reasoning. In Episode 92, with unseen compositions of concepts, the DQN agent fails while NLM-DQN succeeds with the correct Stop decision. Note that in SPF, different ego decisions could significantly affect the city evolution (See Episode 93, Step 125).

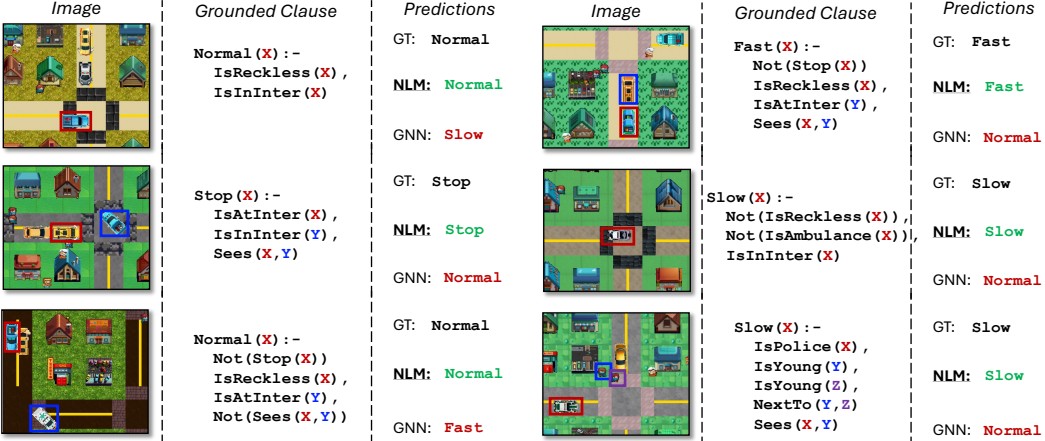

Figure D: Qualitative comparison between NLM [7] and GNN [38] in the hard mode of VAP task. We display the *grounded* clauses, where the involved entities are marked with boxes in corresponding colors. Correct predictions are shown in gree, while the wrong one is in red.

