# OpenReview forum: "LogiCity: Advancing Neuro-Symbolic AI with Abstract Urban Simulation"
_NeurIPS.cc/2024/Datasets_and_Benchmarks_Track — NeurIPS 2024 Track Datasets and Benchmarks Poster_

### Official Review · Reviewer_CmLB · 2024-06-22
**Good work**

**Rating:** 4
**Confidence:** 2
**Clarity:** Fairly yes

**Review:**

The paper is well-structured and addresses a significant gap in NeSy AI research by introducing a flexible and complex simulation environment. However, there are several areas where the paper could be strengthened:

- Experimental Validation: While the experiments cover various methods and scenarios, the evaluation lacks depth in comparing LogiCity's performance against a broader range of benchmarks beyond the ones listed. More comprehensive comparisons with existing NeSy frameworks would solidify the claims.
- Baseline Implementations: The paper implements several baselines but does not provide sufficient details on why certain baselines perform poorly in specific scenarios. A deeper analysis of failure cases would be beneficial.
- Data Representation and Realism: The use of synthetic data for evaluation may limit the generalizability of the findings. Incorporating real-world data or more realistic synthetic data could enhance the validity of the results.

Clarity:
The paper is generally clear but can be improved in a few areas:

- Technical Details: Some sections, particularly those explaining the simulation setup and rule definitions, are dense and could benefit from additional explanatory figures or examples.
- Task Descriptions: The descriptions of the tasks could be more detailed, especially in explaining the differences in complexity and the rationale behind the chosen metrics.

Originality:
The introduction of LogiCity as a benchmark for NeSy AI is original and addresses a clear need in the field. The flexibility of the simulator and the introduction of customizable FOL rules are significant contributions. However, the paper would benefit from a deeper exploration of how these features outperform or enhance existing benchmarks in practical applications.

Significance:
LogiCity has the potential to become a pivotal tool in NeSy AI research, promoting advancements in abstract reasoning and multi-agent interactions. The significance of this work is high, given the current limitations in existing benchmarks. However, the paper needs to provide stronger evidence of the practical impact and scalability of LogiCity in real-world scenarios.

**Strengths:**

- The customizable nature of LogiCity, allowing researchers to adjust concept/rule complexity, is a notable strength. This enables studying adaptation to new abstractions and compositional generalization.
- The inclusion of both sequential decision-making (SPF) and visual reasoning (VAP) tasks provides a more comprehensive evaluation of neuro-symbolic methods.
- The use of first-order logic for defining rules and concepts adds expressiveness compared to propositional logic used in some prior work.
- Empirical results highlight some key challenges for current neuro-symbolic methods, particularly in handling more complex abstractions and long-horizon reasoning.

**Additional Feedback:**

n/a

**Correctness:**

The claims made in the submission are generally correct and supported by the presented experiments. The construction of the benchmark and the design of the evaluation methods are appropriate. However, some concerns need to be addressed:

- Baseline Comparisons: The choice of baselines and their performance should be more critically analyzed. For instance, why certain NeSy methods perform poorly in more complex scenarios should be discussed in detail.
- Statistical Significance: While the results are promising, the paper does not provide sufficient statistical analysis to validate the findings. Including statistical significance tests would strengthen the claims.

**Documentation:**

The documentation for LogiCity appears to be sufficient to support reproducibility.

- Data Collection and Organization: While the paper discusses the synthetic nature of the data, more details on the data generation process and the organization would be helpful.
- Maintenance Plan: The authors should provide a clearer plan for the maintenance and updates of LogiCity, ensuring long-term usability and support for the research community.

**Ethics:**

There are no immediate ethical concerns that stand out in the submission

**Limitations:**

- Conflict Resolution: The authors should provide more detailed strategies or methodologies for resolving rule conflicts and preventing deadlocks in the simulation. This would help in understanding the practical feasibility of their approach.

- Scalability: There is limited discussion on how the system scales with increasing complexity or the number of agents. Addressing this with empirical evidence would strengthen the paper.

**Opportunities For Improvement:**

- Limited novelty in benchmark design: While LogiCity offers more complexity, its core ideas (FOL rules, urban environment) are not fundamentally new. The authors should more clearly articulate what specific novel aspects LogiCity brings beyond increased complexity.
- Lack of theoretical analysis: The paper is heavily empirical, without much theoretical grounding. A formal analysis of the expressiveness or complexity of LogiCity compared to existing benchmarks would strengthen the contribution.
- Narrow range of baseline methods: The evaluation focuses mainly on a few neural and neuro-symbolic methods. Including a broader range of symbolic and hybrid approaches would provide a more comprehensive benchmark.
- Insufficient ablation studies: The paper lacks detailed ablations to isolate the impact of different LogiCity components (e.g. rule complexity, visual rendering). This makes it difficult to understand which aspects are most challenging for current methods.
- Questionable real-world relevance: While more complex than some benchmarks, LogiCity still uses simplified abstract rules. The authors should better justify how insights from LogiCity will transfer to real-world reasoning tasks.
- Lack of human performance baseline: Including human performance on the tasks would provide valuable context for interpreting the AI results.
- Limited discussion of negative results: The paper focuses heavily on positive aspects of LogiCity. A more balanced discussion of potential limitations or failure cases would improve scientific rigor.
- Unclear evaluation metrics: Some metrics (e.g. "Score" in Table 2) are not well-defined. The authors should provide clearer definitions and justifications for their evaluation criteria.
- Rendering approach needs refinement: Using language models and diffusion models for rendering seems unnecessarily complex and potentially introduces unwanted biases. A more controlled procedural rendering approach may be preferable.
- Insufficient comparison to related work: While Table 1 provides a high-level comparison, a more detailed qualitative and quantitative comparison to closely related benchmarks (e.g. CARLA, SCENIC) is needed.

**Relation To Prior Work:**

The paper clearly differentiates LogiCity from previous contributions by emphasizing its flexibility, complexity, and applicability to urban-like environments. The related work section is comprehensive, but there are areas for improvement

**Summary And Contributions:**

This paper introduces LogiCity, a simulator and benchmark for neuro-symbolic AI that models an urban environment using first-order logic (FOL) rules. The key contributions are:

- A customizable FOL-based simulator for urban scenarios
- Two benchmark tasks: Safe Path Following (SPF) for sequential decision-making and Visual Action Prediction (VAP) for one-step reasoning
- Empirical evaluation of several baseline methods on these tasks

The authors claim that LogiCity addresses limitations of existing benchmarks by providing long-horizon reasoning tasks with complex multi-agent interactions and customizable abstractions.

---

> ### Author Rebuttal · Authors · 2024-08-16
>
> ## Reply (3/3)
>
> > *Q8: Definition of Score in Table2*
>
> A8: **We have provided the definition of Score and other metrics in L255, where the "Score" metric is the averaged trajectory return over episodes minus the return of a random agent.** The trajectory step-wise reward has been defined in L193, which involves rule violation, action cost, and overtime punishment.
>
> > *Q9: Differences in task complexity and the rationale behind the chosen metrics*
>
> A9: Due to space limitations, **we have presented the differences in task complexity in Appendix B**, where more complex tasks involve additional concepts and more intricate rules (in terms of longest clause length and rule arity). **The rationale behind the chosen metrics is explained on lines 249-256 for the SPF task and line 289 for the VAP task**.
>
> > *Q10: Negative results of LogiCity.*
>
> A10: The potential negative aspects (limitations) of LogiCity are discussed in the Limitations section, including “probabilistic logic,” “temporal logic,” and the need for a “more continuous map, realistic rendering, and larger action space.” For the failure cases of algorithms, we have provided a deeper analysis **in the Rebuttal PDF**. If you have specific types of “negative results” you would like to see, we would be happy to address those concerns further during our discussion.
>
> > *Q11: Include a broader range of symbolic and hybrid approaches*
>
> A11: We have made every effort to include most of the known symbolic and NeSy methods applicable to LogiCity. **As noted by Reviewer fGrq**, “*The authors acknowledge that most NeSy models are not engineered for the tasks evaluated in their simulation, which is a benefit as it encourages the development of novel approaches to the tasks evaluated by the simulation.*” However, we welcome your suggestions for **any specific additional approaches** that could be relevant during our discussion period.
>
> > *Q12: How to resolve rule conflicts and prevent deadlocks?*
>
> A12: Thank you for this question. Resolving rule conflicts and preventing deadlocks is a longstanding challenge in game design and development, with very few applicable approaches available. Currently, we address conflicts and examine deadlocks by running the simulation 10 times, each involving 1,000 steps, and visualizing the world. As the simulator runs at 10-20 FPS for a 15-20 agent world, **this procedure takes less than an hour and should be affordable for users**.
>
> > *Q13: Including statistical significance tests would strengthen the claims.*
>
> A13: Thanks for this suggestion. Given the limited rebuttal period, we have made every effort to conduct significance tests on key settings and models within the LogiCity benchmark. The **results are presented in Tables 1, 2 of the Rebuttal PDF**.
>
> > *Q14: How does the system scale with increasing complexity or the number of agents? Better to provide stronger evidence of the practical impact and scalability of LogiCity in real-world scenarios.*
>
> A14: Thank you for this question. LogiCity simulation requires only CPUs. In our experiments (SPF), the simulator was running on 32 AMD Ryzen 5950X 16-core CPU processors. **The simulation runs at 20 FPS for a 15-agent world, and approximately 12 FPS for a 20-agent world**, which we believe is sufficient for simulating common real-world scenarios and training RL agents **(See our main paper Table 1)**.
>
> > *Q15: More details on the data generation process*
>
> A15: **We have provided detailed information on data collection in Appendix B**. For SPF tasks, we simply run the ground truth rule-based agent to collect demonstrations. For the VAP task, we sample 100 worlds with random/fixed agent compositions and evolve each of them for 100 steps. Please refer to Appendix B (lines 634-637 and 653-660) for more details.
>
> > *Q16: Long-term Maintenance of LogiCity.*
>
> A16: We are indeed committed to the future development and long-term maintenance of LogiCity. Our current focus includes: (1) integrating multiple lanes, (2) expanding the action spaces of agents, and (3) introducing environmental noise. We also plan to collaborate actively with other research institutions and companies to further enhance LogiCity. However, we want to highlight that **the current version already posed new challenges to notable baselines and introduced valuable insights, opening up research opportunities across various fields.**
>
> > *Q17: Incorporating real-world data or more realistic synthetic data could enhance the validity of the results.*
>
> A17: Thank you for this suggestion. As also recommended by Reviewer p8yp, we plan to distill predicates and rules from the ROAD-R dataset in future versions, bringing the simulation closer to real-life scenarios.
>
> > *Q18: Adding Examples of Rules and Simulator Setup*
>
> A18: Due to space constraints, **we have included the full list of rules, concepts, and their explanations for different modes in our Appendix B, Table A**. We also encourage you to visit our project webpage, where you can explore visual simulations of various rules: [https://jaraxxus-me.github.io/LogiCity/](https://jaraxxus-me.github.io/LogiCity/).  All the necessary information for the simulator setup is also open-sourced on our website.
>
> [1] Chen, Long, et al. "Driving with llms: Fusing object-level vector modality for explainable autonomous driving." 2024 IEEE International Conference on Robotics and Automation (ICRA). IEEE, 2024.

---

> ### Author Rebuttal · Authors · 2024-08-16
>
> ## Reply (2/3)
>
> > *Q4: Using language models and diffusion models for rendering seems unnecessarily complex and potentially introduces unwanted biases.*
>
> A4: Thank you for your concern. We believe that **introducing visual variance is essential**. As shown **in Rebuttal PDF, Table 3**, with visual noise, performance degraded by 10%, and variance increased by 16 times. Additionally, we would like to note that **Reviewer x36W acknowledged that** "*the use of language models and diffusion models for diverse visual rendering is an interesting technical innovation that adds to the simulator's flexibility*." Moving forward, we plan to introduce procedural rendering to provide more photo-realistic images for the VAP task, enhancing the realism of the simulation.
>
> > *Q5: Why do certain baselines perform poorly? Adding insightful failure cases would be helpful.*
>
> A5: Thank you for your suggestion. Due to space constraints in the main paper, we have included the **details of all the baselines in Appendix A** and an in-depth analysis of failure cases in **Appendix G**. For a detailed discussion, please refer to the **Rebuttal PDF**.
> **In Figure 1 (a)**, we highlight that the failure cases from the SPF task primarily concern **compositional generalization**. For instance, the DQN agent failed during testing when encountering a scenario it had not encountered in training, specifically, a scenario with the combination of two pedestrians and an ambulance at an intersection.
> Additionally, **Figure 1 (b) of the Rebuttal PDF** discusses failure cases from the VAP task, which are mainly related to the challenge of distinguishing between different concepts and reasons for action prediction. In LogiCity, for example, reckless cars tend to drive normally at intersections, while other cars proceed more slowly. We observed that the GNN model struggled to differentiate between these concepts, leading to incorrect predictions. We plan to incorporate this insightful analysis and potential future research directions into the main paper in our camera-ready version.
>
> > *Q6: Adding ablation studies to isolate the impact of different simulator components.*
>
> A6: Thank you for raising this important point. Our simulator is built around three key components: rule flexibility, compositional generalization tests, and visual rendering.
>
> - Regarding **rule flexibility**, as illustrated in **Table 1 of our paper**, the performance of models decreases significantly as we move from easy to expert mode, where the rules become more complex. For instance, the TSR of NLM-DQN agents drops from 0.53 to 0.15, which represents a 71.7% reduction.
> - For **compositional generalization**, **Figure 3 in our paper** demonstrates that when using testing agents with different compositions during the testing episodes, instead of the same agents used during training, model performance declines by up to 54%.
> - Lastly, we conducted **additional ablation experiments during the rebuttal period to assess the impact of visual rendering**, specifically by removing the diverse icons generated by a diffusion model. As presented in **Table 3 of the Rebuttal PDF**, the mean results for two models showed a 10% drop in the “with visual variance” tests compared to the “no visual variance” condition. Additionally, the variance in model performance increased 16-fold, indicating that visual variance introduces significant challenges to the task.
>
> In summary, all three components contribute to the difficulty of the tasks, with the greatest impact coming from rule flexibility, followed by compositional generalization, and finally visual rendering.
>
> > *Q7: Adding Theoretical Expressiveness or complexity of LogiCity*
>
> A7: Thanks for this suggestion. We provide Three-fold theoretical expressiveness and complexity of LogiCity (we use the SPF task expert mode for example below).
>
> **State Space:**
>
> The state space in LogiCity's SPF-expert mode is vast. We have 6 different pedestrians, each potentially occupying one of 900 grids, and 8 different cars, each possibly occupying one of 1,400 grids (considering collisions). This results in a total state space of
>  $A^6_{900} \times A^8_{1400} > 10^{39}$. Additionally, the 14 agents can make dynamic action decisions in each step, each with 4 possible actions, leading to **approximately 268 million possible next world states**. In comparison, MiniGrid, a closely related alternative, has a state space of only $\approx$ 16 million and 5 possible next-world states. Thus, LogiCity’s state space is theoretically large enough to provide a much more challenging evaluation.
>
> **Possible plans for completing a worst-case task with every step positive:**
>
> The average (permitted) number of steps to accomplish SPF-expert mode tasks is 281. The longest possible path length is approximately $100$. In each step, the ego agent can do one of the three actions (Stop is neglected here so that each step is ensured making positive progress). To complete a $100$ grid path with at most $281$ steps with three possible actions, the number of possible plans would be $\approx 1.8\times 10 ^{26}$ (solved using dynamic programming). In contrast, considering the 16x16 MiniGrid task, the worst-case scenario is moving from the bottom left to the top right to pick up the key, then moving back to open the door, resulting in the number of plans: $C_{32}^{16} \times C_{32}^{16} \approx 10 ^{18}$. Thus, **there exists a much larger number of plans to complete LogiCity tasks than MiniGrid**, meaning that it provides a scenario with much more possibility.
>
> **Rule Complexity/Expressiveness:**
>
> Given 11 unary and 6 binary predicates, let’s assume the maximum length of the clauses is 6 (which is the case for expert mode) and assume that we have at most 3 arity in the clause. The number of possible clauses would be 11.3 Million (the result is from a program doing the combinatorial search), which we believe is sufficiently large to fulfill user needs.

---

> ### Author Rebuttal · Authors · 2024-08-16
>
> ## Reply (1/3)
>
> The authors appreciate the valuable feedback. We particularly value the positive comments that ***"the customizable nature is a notable strength.", "addresses a significant gap in NeSy AI research", "Empirical results highlight some key challenges"***. We address your specific concerns below:
>
> > *Q1: What novel aspects LogiCity brings beyond complexity? A more detailed qualitative and quantitative comparison to related benchmarks is needed.*
>
> A1: Although CARLA adopts some simple logical rules to control traffic, we respectfully argue that LogiCity is novel in the following aspects:
>
> 1) **Broader Range of Concepts**: LogiCity models a wider variety of concepts compared to driving simulators (**Abstract L9-10, Introduction L56-61**). For example, **LogiCity includes "reckless" cars and "ambulances" and their effects on world dynamics and ego decisions**. These concepts **are inspired by real-world observations and represent a novel approac** compared to CARLA. Specifically, **CARLA includes 5 dynamic concepts that impact ego decisions**: "Vehicle," "Pedestrian," "Traffic Lights," "Signs," and "Dynamic." In contrast, **LogiCity includes 17 relevant concepts** (**detailed in our appendix, Table A**), which are **more fine-grained and thus closer to real-world reasoning**.
> 2) **Expressive Relational (Abstract) Rules**: The semantic and spatial concepts (both unary and binary) are First-Order Logical abstractions (**Abstract L10-13, Introduction L57-61**), which are **Relational** and support **Quantifiers**. This naturally introduced the **abstract reasoning problem**, where algorithms should be **generalizable to any possible compositions** that could be out of training scenarios. **We found most existing methods can't address this very well**.
> 3) **Flexibility in New Concepts and Rule Definition**: The customizable nature of LogiCity allows for the definition of new concepts and rules in a plug-and-play manner, encouraging the development of algorithms that can continuously learn and adapt (**See Abstract L13-15, Introduction L47-50, 62-64**). In contrast, **most existing simulators have fixed dynamics, limiting their ability to support this research direction**.
> 4) **Valid Reasoning Tests**: The richness of abstractions in LogiCity **enables valid real-world reasoning tests for models**, which others fail to provide. GPT-4 can already achieve human-level performance in existing driving simulators \[1\]. However, **LogiCity presents significant reasoning challenges for GPT-4 (See Appendix D, Table C)**, as its scenarios require "in-depth thinking," a capability where humans currently outperform AI (See human baseline in Q3)**. This highlights LogiCity’s potential to advance the next generation of AI algorithms that can intelligently reason and plan, an area where existing simulators are limited.
>
> **These points have been clearly emphasized in our paper**. Moreover, we hope to note that **Reviewer x36W agrees** “*LogiCity presents a novel approach to NeSy AI evaluation by combining customizable FOL rules with an urban environment simulator. This integration is innovative and addresses an important gap in existing benchmarks.*”
>
> > *Q2: How can rules in LogiCity transfer to real-world reasoning scenarios?*
>
> A2: Thanks for this concern. The rules are closely related to real-world scenarios in three key aspects:
>
> 1) **Abstractions for Compositional Generalization**: The rules in LogiCity are abstractions that are **not tied to specific scenes or entities**, enabling **compositional generalization tests**. In the real world, we encounter finite concepts like "ambulance" and "young pedestrian," but their specific combinations are infinitely large and cannot be fully captured in a finite training set. LogiCity captures such a challenge, and as shown in **Figure 3 of our main paper**, **most algorithms struggle with compositional generalization**.
> 2) **Real-World Motivation for Concepts**: The concepts and their effects in LogiCity are **directly inspired by real-world reasoning scenarios**. For instance, reckless cars in LogiCity will not stop at intersections, regardless of their priority. This requires the ego agent to stop carefully at intersections, even if it has a higher priority, to avoid collisions. Most existing simulators have failed to provide such situations.
> 3) **Flexibility and Extensibility:** LogiCity is designed to be flexible and extensible. For example, users can freely introduce a new concept like “Taxi” and define its behavior using FOL with other existing concepts (like pedestrians). We plan to incorporate more realistic rules to further enhance its relevance to real-world scenarios.
>
> > *Q3: Adding human performance baseline.*
>
> A3: Thank you for this suggestion. We fully agree to add a human performance baseline. However, this is non-trivial, as **it requires educating participants on basic logical reasoning and ensuring that each testing answer is a careful decision**. Given the short rebuttal period, we tested a subset of our VAP-hard task, with "Reckless" and "Tiro" cars. The comparative results are presented below:
> | Action/Method | Slow | Normal | Fast | Stop | Overall |
> | :---: | :---: | :---: | :---: | :---: | :---: |
> | **Human** | **95.0** | **92.9** | **48.0** | **83.3** | **81.2** |
> | GPT4 | 40.0 | 54.8 | **28.0** | 30.0 | 40.2 |
> | GPT3.5 | **55.0** | 26.2 | 16.0 | 3.3 | 23.1 |
> | NLM | 40.0 | **69.1** | 0.0 | **50.0** | **44.4** |
> | GNN | 37.3 | 42.5 | 12.5 | 44.4 | 38.5 |
>
> We found that domain experts (NLM) outperform GPT, as the questions require an understanding of LogiCity's abstractions. Interestingly, **after learning from a few in-context examples, humans were able to swiftly grasp the basic abstractions in LogiCity and “generalize” most testing questions, while GPT fell short.** **This demonstrates LogiCity’s potential to advance the next generation of AI models capable of complex abstract reasoning and planning**.

---

> ### Author Response · Authors · 2024-08-23
> **A kind reminder for rebuttal discussion**
>
> Dear reviewer,
>
> The authors sincerely appreciate the time and effort you have already dedicated to reviewing our paper. Following your initial feedback, we have carefully addressed your concerns and provided detailed clarifications and additional experimental results in our rebuttal.
>
> We noticed that you hadn’t got the opportunity to engage further in the discussion after our rebuttal was posted, and we wanted to kindly remind you that we are very open to any additional feedback or questions you may have. We believe that our careful clarification and additional materials may have addressed the issues you raised, and we would be sincerely grateful if you could review our responses and the comments from the other reviewers and consider re-evaluating our submission based on the new information.
>
> Your insights are incredibly valuable to us, and we are eager to ensure that all concerns are fully resolved.
>
> Thank you again for your time and consideration.

---

### Official Review · Reviewer_x36W · 2024-07-20
**LogiCity: A Promising but Limited First Step Towards Comprehensive Neuro-Symbolic AI Evaluation**

**Rating:** 6
**Confidence:** 3

**Review:**

### Quality and Methodology

The work on LogiCity demonstrates a commendable effort in creating a new benchmark for neuro-symbolic AI. The simulator's design shows careful consideration of key challenges in the field, particularly in terms of abstract reasoning and compositional generalization. The experimental methodology is generally sound, with a comprehensive set of baselines tested across varying complexity levels. The authors have thoughtfully chosen evaluation metrics such as Trajectory Success Rate (TSR) and Decision Success Rate (DSR) that provide meaningful insights into system performance.

However, the analysis falls short in several areas. While the experiments cover a broad range of scenarios, there's a notable lack of in-depth examination of failure cases, particularly in more complex settings. This omission represents a missed opportunity to provide valuable insights for future research directions. Additionally, the absence of ablation studies makes it difficult to isolate the impact of different simulator components, limiting our understanding of which features contribute most significantly to the observed results. The statistical rigor of the work could also be improved by including significance tests for performance comparisons.

### Clarity and Presentation

The paper is structured logically and generally well-written, providing a clear explanation of the simulator architecture and its components. The authors have done a commendable job in illustrating key concepts through well-designed figures, particularly Figure 1, which effectively summarizes the LogiCity framework. The detailed descriptions of experimental setups and baseline configurations are helpful for understanding the methodology.

Nevertheless, the presentation has room for improvement. Many technical details crucial for a full understanding of the work are relegated to appendices, which may hinder readers' comprehension of the main text. The distinction between different complexity modes (easy, medium, hard, expert) could be articulated more clearly. The paper would benefit from more concrete examples of specific rules and their implications in the simulator, which would enhance reader understanding of the system's capabilities and limitations.

### Originality and Innovation

LogiCity presents a novel approach to neuro-symbolic AI evaluation by combining customizable first-order logic (FOL) rules with an urban environment simulator. This integration is innovative and addresses an important gap in existing benchmarks. The approach to testing compositional generalization by varying agent sets is particularly noteworthy and represents a valuable contribution to the field.

The use of language models and diffusion models for diverse visual rendering is an interesting technical innovation that adds to the simulator's flexibility. However, it's important to note that the basic concept of an urban environment simulator is not new, with existing platforms like CARLA and SUMO already established in the field. Many of the baseline methods and evaluation metrics used are standard in the field, which somewhat limits the overall originality of the work.

### Significance and Impact

LogiCity has the potential to make a significant contribution to the field of neuro-symbolic AI. It addresses important gaps in existing benchmarks, particularly in terms of flexibility and customization. The platform provides a foundation for the systematic study of abstract reasoning and generalization, which are crucial challenges in AI research. The potential applications in real-world domains like autonomous driving lend practical relevance to the work.

However, the current design of LogiCity may be too constrained to fully capture the complexity of real-world reasoning tasks. The focus on a simplified urban environment, while providing a concrete scenario for testing, may limit broader applicability to other domains. The long-term impact of LogiCity will largely depend on its adoption by the research community and continued development to address its current limitations.

Overall, while LogiCity represents a valuable step forward in neuro-symbolic AI evaluation, it should be viewed as a promising starting point rather than a definitive solution. The simulator provides a flexible foundation for studying important aspects of AI reasoning, but its simplified environment and rule structure may not fully capture the complexities of real-world scenarios. As the platform evolves to address these limitations, it has the potential to become a standard benchmark in the field, driving progress in abstract reasoning, compositional generalization, and the integration of symbolic knowledge with deep learning.

**Strengths:**

The LogiCity submission presents several notable strengths that contribute to its significance and relevance to the broader research community.

Firstly, the work addresses a crucial gap in existing neuro-symbolic AI benchmarks. By providing a flexible, customisable environment that supports both long-horizon sequential decision-making and visual reasoning tasks, LogiCity offers a more comprehensive platform for evaluating AI systems. This breadth of capability is particularly relevant as the field moves towards more integrated approaches that combine symbolic reasoning with deep learning.

The simulator's design demonstrates a thoughtful consideration of key challenges in neuro-symbolic AI. The ability to customise concepts, rules, and agent sets allows researchers to systematically study different aspects of reasoning and generalization. This flexibility is a significant strength, as it enables the incremental increase of task complexity and the testing of AI systems across various dimensions of abstraction and reasoning.

The focus on compositional generalization, achieved through varying agent sets, is particularly innovative and addresses a critical aspect of AI development. The capacity to apply learned rules to new entity combinations is crucial for creating more robust and adaptable AI systems, making this feature of LogiCity especially relevant to the research community.

The quality of the research is evident in the comprehensive evaluation framework presented. The authors have conducted thorough experiments across multiple baseline methods and complexity levels, providing a nuanced view of the strengths and weaknesses of different AI approaches. The choice of evaluation metrics, such as Trajectory Success Rate and Decision Success Rate, offers meaningful insights into system performance.

From an ethical and social perspective, LogiCity's focus on an urban environment scenario has clear connections to real-world applications like autonomous driving. This practical relevance enhances the potential societal impact of the research. Moreover, by providing an open-source platform for studying abstract reasoning and decision-making in complex environments, the work contributes to the development of more transparent and interpretable AI systems – a crucial consideration for ethical AI development.

The integration of visual reasoning through the Visual Action Prediction task is another significant strength. This bridges the gap between symbolic reasoning and real-world perception, addressing a key challenge in creating AI systems that can operate effectively in complex, real-world environments.

Lastly, the authors' decision to open-source the simulator and datasets represents a valuable contribution to the research community. This openness not only promotes reproducibility but also has the potential to accelerate progress in neuro-symbolic AI by providing a common benchmark for comparison and collaboration.

In summary, LogiCity's strengths lie in its comprehensive and flexible approach to neuro-symbolic AI evaluation, its focus on critical challenges like compositional generalization, the quality of its experimental design, and its potential to drive progress in developing more robust, interpretable, and ethically-aligned AI systems. While there are areas for improvement, the work represents a significant step forward in the field and offers a promising foundation for future research in neuro-symbolic AI.

**Additional Feedback:**

LogiCity presents a promising new benchmark for neuro-symbolic AI, offering a flexible and customizable urban environment for evaluating complex reasoning tasks. The authors have done commendable work in designing a simulator that addresses key challenges in the field, particularly in testing compositional generalization and integrating visual reasoning with symbolic logic. The comprehensive evaluation across multiple baselines and complexity levels provides valuable insights into the current capabilities and limitations of various AI approaches.

To further strengthen this excellent contribution, the authors could consider expanding their analysis of failure cases and including ablation studies to isolate the impact of different simulator components. A more detailed comparison with existing autonomous driving simulators would help contextualize LogiCity's specific advancements. Additionally, discussing plans for maintaining LogiCity's relevance as the field progresses and addressing potential biases in rule design would enhance the paper's impact.

Looking forward, it would be interesting to explore how LogiCity could be extended to incorporate more advanced logical constructs or evaluate ethical decision-making in AI systems. The authors' insights on how performance in LogiCity might translate to real-world scenarios would also be valuable. Overall, LogiCity represents a significant step forward in neuro-symbolic AI evaluation, and with some refinements, it has the potential to become a standard benchmark in the field, driving progress in abstract reasoning and the integration of symbolic knowledge with deep learning.

**Clarity:**

The paper is generally well-written, but there are areas where clarity and presentation could be improved:

Strengths:

1. Structure: The paper follows a logical structure, introducing the concept, detailing the simulator, explaining the tasks, and presenting the experiments.

2. Technical explanations: The authors provide clear explanations of the simulator architecture and components, particularly in Section 3.

3. Visual aids: Figures, especially Figure 1, effectively illustrate key concepts of LogiCity.

4. Experimental setup: The descriptions of experimental setups and baseline configurations are detailed and helpful.

Areas for improvement:

1. Technical detail distribution: Some crucial technical details are relegated to appendices, which can make it challenging to fully understand the main text without frequent referencing.

2. Complexity modes: The distinction between different modes (easy, medium, hard, expert) could be explained more clearly in the main text.

3. Rule examples: More concrete examples of specific rules and their implications in the simulator would enhance understanding of the system's capabilities and limitations.

4. Abbreviations: Some technical terms and abbreviations are used without full explanation on first use, which might confuse readers less familiar with the field.

5. Balancing detail: In some sections, there's an imbalance between high-level description and technical detail. Finding a better balance could improve overall readability.

6. Transitions: Some sections could benefit from improved transitions to better connect different parts of the paper.

7. Abstract and introduction: These sections could be tightened to more concisely convey the key contributions and significance of the work.

8. Discussion of limitations: While limitations are mentioned, a more comprehensive discussion integrated into the main body of the paper would provide a more balanced presentation.

Overall, while the paper is competently written and generally clear, there is room for improvement in terms of presentation, balance of detail, and overall clarity. Addressing these issues would make the paper more accessible and impactful for a broader audience within the AI research community.

**Correctness:**

The claims made in the submission regarding LogiCity appear to be generally correct, and the benchmark is constructed in a thoughtful manner. However, there are some areas where the evaluation methods and experiment design could be improved or clarified:


Correctness of claims:
The main claims about LogiCity's features and capabilities seem to be accurately represented. The simulator does indeed provide a customizable urban environment with first-order logic rules, support for both sequential decision-making and visual reasoning tasks, and the ability to test compositional generalization.


Benchmark construction:
The overall design of LogiCity as a benchmark is sound. The authors have put considerable thought into creating a flexible environment that can be customized to test different aspects of neuro-symbolic AI. The inclusion of both the Safe Path Following (SPF) and Visual Action Prediction (VAP) tasks provides a good range of challenges.


Evaluation methods and experiment design:
While generally appropriate, there are some areas where the evaluation could be strengthened:

Baseline selection: The authors have included a good range of baselines, covering symbolic, neuro-symbolic, and pure neural approaches. This provides a comprehensive view of current capabilities.


Metrics: The use of Trajectory Success Rate (TSR), Decision Success Rate (DSR), and weighted accuracy are appropriate for evaluating performance on the tasks.


Complexity levels: The inclusion of easy, medium, hard, and expert modes allows for a systematic evaluation of AI capabilities across different levels of reasoning complexity.


Compositional generalization: The approach of using different agent configurations for training and testing is a sound way to evaluate compositional generalization.

Areas for improvement:

Statistical significance: The paper would benefit from including statistical significance tests for the performance comparisons between different methods.
Ablation studies: The authors could strengthen their evaluation by including ablation studies to isolate the impact of different simulator components or rule complexities.


Failure analysis: A more in-depth analysis of failure cases, especially in complex scenarios, would provide valuable insights.
Reproducibility: More details on the computational requirements and specific configurations used for experiments would enhance reproducibility.


Comparison to related benchmarks: A more explicit comparison to existing urban environment simulators or autonomous driving benchmarks would help contextualize LogiCity's contributions.
Long-term stability: The paper could address how the benchmark will be maintained and updated over time to remain relevant as the field progresses.

**Documentation:**

Regarding LogiCity, which is primarily presented as a benchmark and simulator rather than a traditional dataset, the authors have provided a substantial amount of detail to support reproducibility. However, there are some areas where additional information would be beneficial:

Data collection and organization:
The paper provides a good overview of how the simulator generates data, including the use of first-order logic rules and the rendering process for visual data. However, more specific details on the generation of urban layouts and agent behaviours could be included.

Availability and maintenance:
The authors state that "All the code and data are open-sourced at our website." However, the specific URL is not provided in the paper (likely due to anonymity requirements for the review process). A clear statement on long-term maintenance plans for the simulator and associated datasets would be valuable.

Ethical and responsible use:
The paper briefly mentions potential negative societal impacts, specifically the sim-to-real gap. However, a more comprehensive discussion of ethical considerations and guidelines for responsible use of the benchmark would strengthen the submission.

Documentation and intended uses:
While the paper describes the intended uses of LogiCity for evaluating neuro-symbolic AI systems, a separate, more detailed documentation of the simulator's features, limitations, and best practices for use would be beneficial.

Reproducibility:
The paper provides a good level of detail on the simulator's architecture, tasks, and evaluation metrics. The appendices include additional information on baseline configurations and experimental setups. However, some aspects could be improved:

1. More specific details on computational requirements for running the simulator and experiments.
2. Clear instructions for setting up and running the simulator.
3. Information on any random seeds or other factors that might affect reproducibility.
4. Details on the specific versions of dependencies and libraries used.

Hosting, licensing, and maintenance plan:
While the authors mention that the code and data are open-sourced, specific information on the hosting platform, licensing terms, and long-term maintenance plans are not provided in the current version of the paper.

In conclusion, while the authors have provided a good foundation for reproducibility and understanding of the benchmark, there are several areas where additional information and documentation would be valuable. Addressing these points would significantly enhance the usefulness and accessibility of LogiCity for the broader research community.

**Ethics:**

Based on the information provided in the submission, there do not appear to be major ethical concerns with LogiCity that would warrant significant further review. However, there are a few areas that merit further discussion and clarification:

1. Data privacy, copyright, and consent:
As LogiCity is a simulated environment, there are no direct concerns about data privacy or consent from human subjects. However, the authors should clarify the licensing terms for the simulator and any generated datasets to ensure proper attribution and use.

2. Data quality and representativeness:
The simplified urban environment in LogiCity may not fully represent the complexity and diversity of real-world urban scenarios. The authors should provide a more detailed discussion on the limitations of this representation and potential biases that might be introduced.

3. Discrimination, bias, and fairness:
While not explicitly discussed in the paper, there is potential for bias in the hand-designed rules and agent behaviours. The authors should address how they've attempted to mitigate potential biases and discuss any limitations in this regard.

4. Safety and security:
As LogiCity is intended for use in developing AI systems that could potentially be applied to real-world scenarios (e.g., autonomous driving), the authors should provide guidelines on responsible use and emphasize the limitations of transferring results from the simulator to real-world applications.

5. Environmental Impact:
The paper does not discuss the computational resources required to run LogiCity or conduct experiments. Given the increasing focus on the environmental impact of AI research, the authors should provide some information on the energy consumption and computational requirements of their simulator.

6. Human rights (including surveillance):
While LogiCity itself does not appear to raise direct human rights concerns, the authors should discuss potential implications of using such simulated environments for developing AI systems that could be used in urban surveillance or monitoring.

These points do not represent severe ethical concerns but rather areas where additional discussion and clarification would be beneficial. The authors should be encouraged to address these points to ensure responsible development and use of their benchmark.

Overall, the ethical considerations for LogiCity appear to be manageable and in line with typical concerns for AI research tools and benchmarks. However, a more comprehensive discussion of these issues in the paper would strengthen the submission and demonstrate the authors' commitment to ethical AI research.

**Limitations:**

The authors have made some effort to address limitations and potential negative societal impacts of their work, but there are several areas where this discussion could be expanded and improved. Here's an assessment of their approach and suggestions for improvement:

Addressed limitations:
• The authors acknowledge that LogiCity is a first step and that more complex scenarios still pose significant challenges for current methods.
• They mention the need for users to pre-define rule sets that are conflict-free and do not cause deadlocks.
• The paper notes that LogiCity currently does not support temporal logic.

Addressed potential negative impacts:
• The authors briefly mention that rules in LogiCity may not accurately reflect real life, introducing a sim-to-real gap.

Areas for improvement:

1. Expanded discussion of limitations:
   • Provide a more detailed analysis of the simulator's limitations in representing real-world complexity.
   • Discuss the potential for overfitting to the specific urban environment and rule structures in LogiCity.
   • Address the challenge of scaling the simulator to more complex scenarios and larger agent sets.

2. Deeper exploration of potential biases:
   • Examine how the hand-designed rules might introduce biases into the evaluation process.
   • Discuss strategies for identifying and mitigating these biases.

3. Broader societal impact analysis:
   • Consider the potential misuse of systems trained on LogiCity in real-world applications.
   • Discuss the implications of using simplified urban models for developing AI systems that may be deployed in complex, real-world environments.

4. Ethical considerations:
   • Address the ethical implications of developing AI systems that make decisions in urban environments, particularly regarding safety and fairness.
   • Discuss how LogiCity could be used to evaluate the ethical decision-making capabilities of AI systems.

5. Environmental impact:
   • Include a discussion on the computational resources required to run LogiCity and its potential environmental impact.
   • Suggest strategies for minimising this impact in future iterations.

6. Long-term research directions:
   • Provide a more detailed roadmap for addressing current limitations in future work.
   • Discuss how the community can contribute to expanding and improving LogiCity.

7. Comparison to real-world scenarios:
   • Include a more thorough discussion of how performance in LogiCity might translate to real-world scenarios, and the potential risks of overestimating AI capabilities based on simulator performance.

By addressing these points, the authors would provide a more comprehensive and balanced view of LogiCity's limitations and potential impacts. This would not only improve the paper but also demonstrate a commitment to responsible AI development and evaluation.

**Opportunities For Improvement:**

In terms of significance and relevance, the primary limitation lies in the simulator's current level of environmental complexity. Whilst the urban grid environment represents a step forward from more abstract benchmarks, it may still be too simplistic to capture the full intricacy of real-world reasoning scenarios. This constraint potentially limits the broader applicability of insights gained from LogiCity to more complex, real-world AI challenges. The focus on a specific urban setting, while relevant to domains like autonomous driving, may not fully address the needs of researchers working in other areas of AI application.


The quality of the research, while generally good, has some notable shortcomings. There is a lack of in-depth analysis of failure cases, particularly in more complex scenarios. This omission represents a missed opportunity to provide valuable insights for future research directions. Additionally, the absence of ablation studies makes it difficult to isolate the impact of different simulator components, limiting our understanding of which features contribute most significantly to the observed results. The statistical rigour of the work could also be improved by including significance tests for performance comparisons.


From a methodological perspective, the current rule set, while customisable, may not capture the full range of logical relationships needed for comprehensive neuro-symbolic AI evaluation. The lack of support for more advanced logical constructs, such as temporal logic or probabilistic rules, could limit the simulator's ability to represent and evaluate more sophisticated reasoning capabilities.
Ethically and socially, there are concerns about potential biases in the hand-designed rules. While the flexibility to customise rules is a strength, it also introduces the risk of inadvertently embedding biases or oversimplifications into the evaluation framework. This could lead to skewed assessments of AI systems or the development of systems that perform well in LogiCity but fail to generalise to real-world scenarios with more nuanced and complex rule structures.


The work also falls short in providing a thorough comparison to existing domain-specific benchmarks, particularly in the realm of autonomous driving and urban environment simulation. This lack of contextualisation makes it challenging to fully appreciate the specific advancements that LogiCity brings to the field.


From a practical standpoint, the paper provides limited information on the computational requirements for running the simulator and experiments. This omission could impact reproducibility and accessibility, particularly for researchers with limited computational resources.
Lastly, while the open-source nature of LogiCity is commendable, the long-term impact and significance of the work will heavily depend on its adoption by the research community and continued development to address its current limitations. Without sustained effort to evolve the platform, there's a risk that its relevance could diminish over time as the field advances and new challenges emerge.

**Relation To Prior Work:**

The paper makes an effort to distinguish LogiCity from previous contributions, primarily through a comparative table (Table 1) and by identifying specific gaps in existing benchmarks. The authors highlight LogiCity's unique features, such as customizable first-order logic rules and the ability to test compositional generalization by varying agent sets. They also emphasize the simulator's capacity for long-horizon reasoning tasks with complex multi-agent interactions, which they argue is lacking in existing benchmarks.

However, the differentiation from previous work could be strengthened in several ways. While the paper mentions other benchmarks and simulators, it lacks in-depth comparisons of specific capabilities and limitations. Given the urban environment context, a more thorough comparison with existing autonomous driving simulators would be beneficial. The authors could also more explicitly state which aspects of LogiCity are entirely novel versus improvements on existing ideas.

To further improve this aspect, the paper would benefit from a brief overview of the evolution of NeSy AI benchmarks to help readers better understand LogiCity's place in the field's progression. Where possible, including quantitative comparisons with existing benchmarks would strengthen the differentiation. Additionally, the authors could more clearly articulate how LogiCity opens up new research directions that were not possible with previous benchmarks. Overall, while the paper does address how LogiCity differs from previous contributions, this aspect could be expanded and made more explicit to better highlight its significance in the context of NeSy AI research.

**Summary And Contributions:**

This submission introduces LogiCity, a new simulator and benchmark for neuro-symbolic AI research. The key contributions are:

1. A customizable urban environment with first-order logic (FOL) rules governing agent behaviours.
2. Support for both long-horizon sequential decision-making and visual reasoning tasks.
3. Ability to test compositional generalization by varying agent sets.
4. Flexibility to adjust reasoning complexity through customizable concepts, rules, and agent configurations.

The authors present two main tasks:
- Safe Path Following (SPF): A sequential decision-making task
- Visual Action Prediction (VAP): A single-step visual reasoning task

Experiments compare various baseline methods, including symbolic, neuro-symbolic, and pure neural approaches. Results show that while neuro-symbolic methods generally outperform others, they still struggle with more complex scenarios.

The paper demonstrates that LogiCity can be used to:
1. Evaluate the ability of AI systems to learn abstract rules
2. Test generalization to unseen agent compositions
3. Assess performance on increasingly complex reasoning tasks

Overall, LogiCity aims to address gaps in existing neuro-symbolic AI benchmarks by providing a flexible environment for studying abstract reasoning in a simplified urban setting.

---

> ### Author Rebuttal · Authors · 2024-08-16
>
> ## Reply (3/3)
>
> > *Q8: Computational requirements and the challenge of scaling for running LogiCity*
>
> A8: **We have provided the hardware details used in our experiments on lines 235 and 278**. For SPF, we used a single NVIDIA RTX 3090 Ti GPU with 32 AMD Ryzen 5950X 16-core processors, and for VAP, we trained all models using a single NVIDIA H100 GPU. LogiCity’s simulation is very lightweight and only requires CPUs, **running at 20 FPS for a 15-agent world**, which we believe is sufficient for simulating common scenarios and RL training. The primary scaling challenge lies in rendering speed, which is not the main focus of our work and is currently completed offline after the simulation. In the future, we plan to incorporate graphical rendering to enable photo-realistic simulation.
>
> > *Q9: Continued future development and long-term maintenance plan of LogiCity*
>
> A9: Thank you for this comment. We are indeed committed to the continued development and long-term maintenance of LogiCity. Specifically, we are currently working on (1) incorporating more realistic maps with multiple lanes, (2) augmenting the action spaces of agents, and (3) introducing environmental noise. **Our code is well documented and has been open-sourced**, we are actively collaborating with other research institutions and companies to expand LogiCity further. However, we would like to emphasize that **the current version is already challenging and insightful, having opened up research opportunities across multiple fields.**
>
> > *Q10: More concrete rule examples and differences between each mode in LogiCity.*
>
> A10: Thanks for this suggestion. Due to the limited space, **we have provided the full list of rules and concepts in different modes in our Appendix B**. We also invite you to review our project webpage (https://jaraxxus-me.github.io/LogiCity/) and visualize the simulation examples of different rules at the "Simulation Visualization" Section.
>
> > *Q11: Open-source LogiCity (project website)*
>
> A11: **We have fully open-sourced LogiCity** at: [https://jaraxxus-me.github.io/LogiCity/](https://jaraxxus-me.github.io/LogiCity/)
>
> \[1\] Chen, Long, et al. "Driving with llms: Fusing object-level vector modality for explainable autonomous driving." 2024 IEEE International Conference on Robotics and Automation (ICRA). IEEE, 2024\.

---

> > ### Comment · Reviewer_x36W · 2024-08-22
> >
> > I am happy with these responses

---

> ### Author Rebuttal · Authors · 2024-08-16
>
> ## Reply (2/3)
>
> > *Q5: Potential biases may be introduced in the hand-designed rules, which could make methods overfit and fail to generalize to real-world.*
>
> A5: Thanks for raising this concern. We agree that the current rules in LogiCity are “abstractions” motivated by the real world, and we are committed to further developing and extending the simulator.
> However, we would like to emphasize that:
>
> 1) The rules in LogiCity **are closer to real-world scenarios compared to existing simulators**. LogiCity models a broader range of fine-grained concepts and their impact on world dynamics and ego decisions. [Here](https://openreview.net/forum?id=M32Ldpp4Oy#discussion:~:text=Ambulance%22%20cars.-,The%20reckless%20cars,-drive%20faster%20when) is a concrete example for “reckless” cars.
> 2) LogiCity is currently designed to evaluate the learning of "abstract" rules **applicable to the real world**. It serves as a platform for developing and validating the capabilities of **learning algorithms**, such as DQN and NLM. We plan to extend and validate LogiCity further to support the training of "deployable models."
> 3) According to our extensive experiments, most existing algorithms struggle to learn "abstract" rules effectively. This suggests that LogiCity can ***push the development of abstract reasoning/decision-making AI algorithms forward before they are ready for real-world deployment***.
>
> Therefore, while we acknowledge that the current hand-designed rules may introduce some bias and are a simplified version of real-world scenarios, we believe that (1) LogiCity is **closer to the real world** than existing simulators, (2) it is **instrumental in AI algorithm development and validation**, and (3) it **presents challenges that most existing algorithms currently find difficult** to overcome before deployment consideration.
>
> > *Q6: Discussion/Comparison of Autonomous Driving simulators like CARLA and SUMO*
>
> A6: We agree that existing simulators like CARLA and SUMO are also focused on urban simulation. However, LogiCity is **significantly different and presents more challenging reasoning tasks**:
>
> Qualitatively, LogiCity models a broader range of concepts and their effects on the environment. For example, LogiCity includes "reckless cars," "ambulances," and "police cars," which **are often overlooked in autonomous driving simulators**. Reckless cars in LogiCity drive faster near intersections, increasing their chances of arriving first. Agents must stop and give way when they encounter an ambulance. We invite the reviewer to visit our website for more examples.
>
> Quantitatively, with the diverse concepts and abstract rules, LogiCity significantly raised the reasoning challenge. **GPT4 can be used to generate ground-truth labels for autonomous driving questions** \[1\], which is close to a human expert. However, **it falls far behind humans in LogiCity as shown below**, demonstrating the reasoning challenge from LogiCity.
>
> | Action/Method | Slow | Normal | Fast | Stop | Overall |
> | :---: | :---: | :---: | :---: | :---: | :---: |
> | **Human** | **95.0** | **92.9** | **48.0** | **83.3** | **81.2** |
> | GPT4 | 40.0 | 54.8 | **28.0** | 30.0 | 40.2 |
> | GPT3.5 | **55.0** | 26.2 | 16.0 | 3.3 | 23.1 |
> | NLM | 40.0 | **69.1** | 0.0 | **50.0** | **44.4** |
> | GNN | 37.3 | 42.5 | 12.5 | 44.4 | 38.5 |
>
> > *Q7: Quantitative comparisons/More Explanation of the Difference between LogiCity and existing NeSy benchmarks.*
>
> A7: Thank you for this suggestion. We quantitatively compare LogiCity with MiniGrid (for sequential decision-making tasks) and VisualSudoku (for visual reasoning tasks) below:
>
> MiniGrid: **As shown in Rebuttal PDF Figure 2**, MiniGrid is a 16x16 grid world with (possibly) one key, one door, and an agent. The agent must pick up the key and open the door to complete the task. The largest state space is approximately $A^3_{256} \approx 16.6$ million. Additionally, MiniGrid has only one dynamic (ego) agent, leading to 5 possible next world states given 5 possible ego actions. In contrast, LogiCity's SPF-expert mode includes 6 different pedestrians (potentially occupying 900 grids) and 8 different cars (potentially occupying 1400 grids, considering collisions). The state space for LogiCity is $A^6_{900} \* A^8_{1400} > 10^{39}$, which is several orders of magnitude larger, can already be considered a continuous problem. Moreover, LogiCity features 14 dynamic agents, each capable of 4 possible actions, resulting in approximately 268 million possible next-world states, to which MiniGrid is not comparable.
>
> VisualSudoku: **Please see our Rebuttal PDF Figure 2**, It is a 9x9 sudoku board with Minist images as input. The image space between VisualSudoku and LogiCity VAP is similar, we compare their symbolic space below. There exist approximately $6.67\times10^{21}$ possible sudoku boards. While for LogiCity VAP hard mode, we have 6 pedestrians (3 possible types) and 8 cars (6 possible types), each pair of them could have 6 possible binary relationships. This results in the number of possible symbolic scenarios:
> $(3^6) \times (6^8) \times ( 2^6)^{14\times13} \approx 3.98 \times 10^{392}$, which is much larger than Sudoku. Besides, LogiCity features FOL, which means **the number of agents can be arbitrarily large** (while sudoku is based on propositional logic and is thus fixed to $9\times9=81$).
> As conclusion, Compared with existing NeSy benchmarks, LogiCity has a **theoratically much larger state space (nearly continuous) and supports scenarios with arbitrary agent composition**, which is closer to our real life. We will provide these detailed comparisons in our supplementary material.

---

> > ### Comment · Reviewer_x36W · 2024-08-22
> >
> > I am happy with this response from the authors as they answer key concerns

---

> > ### Comment · Reviewer_x36W · 2024-08-22
> >
> > The additional comparisons and statistics support the argument for the potential of logi city, and its ability to extend beyond current alternatives. Obviously this does rely on future development, which includes the potential to address concerns around rule introduced bias.

---

> > > ### Author Response · Authors · 2024-08-22
> > > **Thank you for your reply**
> > >
> > > We are very pleased that our response addressed your key concerns and appreciate your positive feedback. We are delighted that you recognize LogiCity's potential to extend beyond current alternatives. Regarding rule-introduced bias, we are actively collecting more realistic rules to incorporate. However, we would like to emphasize that the LogiCity benchmark **has already introduced new challenges (even the most recent GPT-4 struggles with them)**, and the simulator **provides unique features (such as flexible first-order abstractions) that are not currently supported by others**.
> > >
> > > We are sincerely grateful that you provided a positive initial rating to support the acceptance of our work. As we are having a very constructive discussion, we would be more than happy if you could consider providing a slightly stronger final rating.
> > >
> > > Thank you for your thoughtful review and consideration.

---

> ### Author Rebuttal · Authors · 2024-08-16
>
> ## Reply (1/3)
>
> We sincerely appreciate your thorough review and comments. We are pleased to see your positive comments: "***addresses a crucial gap in existing neuro-symbolic AI benchmarks***", "***demonstrates a thoughtful consideration of key challenges in neuro-symbolic AI***", "***is particularly innovative and addresses a critical aspect of AI development***", "LogiCity's focus on an urban environment scenario ***has clear connections to real-world applications*** like autonomous driving", and that “***The platform provides a foundation for the systematic study of abstract reasoning and generalization***”. We address your specific concerns below:
>
> > *Q1: LogiCity should be viewed as a promising starting point rather than a definitive solution.*
>
> A1: Thank you for raising this concern. We fully agree that LogiCity represents a promising starting point, and extending it to better reflect real-world complexity is indeed important. We are committed to the ongoing development and maintenance of the code library and are actively engaging in collaborations with multiple research institutions and companies to enhance the simulator. However, we would like to emphasize that:
>
> 1) LogiCity is currently the **first simulator based on customizable First-Order Logic (FOL)** that **supports realistic multi-agent interactions**, which is **a novel contribution to the community**.
> 2) The primary focus of LogiCity is **the complexity of "abstract reasoning,"** requiring models to generalize compositionally in environments with multiple dynamic agents. This challenge, inspired by real-world complexity, **is not well addressed by other benchmarks, making LogiCity’s challenges particularly insightful**.
> 3) Due to its novel and insightful design, LogiCity is already **highly challenging for most existing algorithms (as shown in Table 2\)**, opening up numerous research opportunities for the development of more advanced methods.
>
> Therefore, while we agree with the importance of continued development, we respectfully emphasize that **it is novel, insightful, and challenging**, which are valuable features for an impactful benchmark.
>
> > *Q2: Adding analysis of in-depth examinations of failure cases*
>
> A2: Thank you for this suggestion. Due to space constraints in the main paper, we have included **in-depth analysis of failure cases in the Appendix G.**
>
> **In the Rebuttal PDF, Figure 1 (a)**, we illustrate that **the failure cases from the SPF task primarily involve compositional generalization**. For example, in Episode 92, the DQN agent failed during testing because it had not encountered a scenario during training where two pedestrians were at an intersection along with an ambulance.
> **In the Rebuttal PDF Figure 1 (b)**, we show that **the failure cases from the VAP task mainly involve difficulty in distinguishing between different concepts or reasons for action prediction**. For instance, in LogiCity, reckless cars drive normally through intersections, while other cars slow down. We found that the GNN model struggled to distinguish between these concepts, leading to incorrect predictions.
>
> > *Q3: Including significance tests*
>
> A3: Thanks for this suggestion. Due to the short rebuttal period, we have tried our best to implement significant tests on several representative settings and models in the LogiCity benchmark. **The results are shown in the Rebuttal PDF, Tables 1, and 2**. We will incorporate the full tests in our camera-ready version.
>
> > *Q4: Adding ablation studies to isolate the impact of different simulator components.*
>
> A4: Thanks for this comment. Our simulator has three key components: rule flexibility, compositional generalization tests, and visual rendering.
>
> - Rule Flexibility: **As shown in Table 1 of our paper**, moving from easy to expert mode (which involves more complex rules) significantly decreases model performance. For instance, the TSR of NLM-DQN agents drops from 0.53 to 0.15, a 71.7% decline.
> - Compositionality: **As illustrated in Figure 3 of our paper**, when using testing agents (with different compositions) instead of training agents in the testing episodes, model performance drops by up to 54%.
> - Visual Rendering: During the rebuttal period, we conducted ablation studies on the diverse icons generated by a diffusion model. The results, **shown in Rebuttal PDF Table 3**, indicate that **introducing visual variance leads to a 10% drop in the mean results of two models and a 16-fold increase in the variance** of their results, demonstrating the added challenge posed by visual variance.
>
> We will incorporate these additional results in the final version of our paper.

---

### Official Review · Reviewer_fGrq · 2024-07-21
**A well designed framework and dataset for evaluating and advancing neuro-symbolic AI**

**Rating:** 8
**Confidence:** 4

**Review:**

This work presents a novel benchmark for the evaluation of NeSy agents op-
erating in urban environments. They claim that existing approaches for the
evaluation of such agents is limited in that such approaches fail to evaluate
long-horizon reasoning tasks with complex multi-agent interactions. This claim
is supported by a review of existing benchmarks and their features. Further-
more, the experiments presented in this paper support the authors claim that
due to the lack of benchmarks available for this type of task, existing NeSy
agents generally perform poorly within their simulation. In the Safe Path Fol-
lowing task, no model exceeds a trajectory success rate ≥ 50% in the ”hard”
cateogry. Similarly, for the Visual Action Prediction task, the highest reported
weighted accuracy among all evaluated models is 54% in the ”hard” category.
These results support the authors claim that their benchmark poses a signifi-
cant challenge to existing approaches in sophisticated reasoning scenarios, but
encourages the gradual development of new NeSy agents due to the flexibility
of their environment.

**Strengths:**

The authors introduce a new simulation for the evaluation of NeSy urban agents.
They state that their simulation raises the abstract reasoning complexity with
long-horizon multi-agent scenarios, which are not adequately addressed by cur-
rent methods. This claim is supported in their experimental evaluation. Their
environment is flexible, in that users can configure their own concepts, rules, and
agents. This design choice allows users to develop agents for various scenarios
in diverse urban environments.

**Additional Feedback:**

See other sections.

**Clarity:**

The quality of writing in the paper is adequate, with occasional deviations from
my understanding of the style of academic writing. Some examples include:
• “While we show that existing NeSy approaches [7, 11] perform better in
learning abstractions, both from scratch and continually, more complex
scenarios from LogiCity still pose significant challenges for prior arts [7,
10, 11, 32–38]. On the one hand, LogiCity raises the abstract reasoning
complexity with long-horizon multiple agents scenario, which have not
been adequately addressed by current methods.” (questionable term “prior
arts”, is “scenario” meant to be plural? If not, should be “... the long-
horizon multiple agents scenario.”)
• “LogiCity allows for flexible configuration on the concepts and FOL rules...”
(line 314, should say “of” rather than “on”)
• “We present two modes for VAP task...” (line 285, should say “the VAP
task”)
• “The Safe Path Following (SPF) task aims at evaluating sequential decision-
making capability...” (line 178, ”aims to evaluate” is more concise.)
• “During test, the models are expected to predict the actions for a different
set of agents.” (line 216, should say “During testing”)
• “Additionally, NeSy framework [7, 34] outperform pure neural agents [34,
36] by finding better...” (line 263, is framework meant to be plural?)

Aside from the examples of non-standard phrasing or deviant grammar,
the conceptual clarity of the work is satisfactory.

**Correctness:**

The authors acknowledge that most NeSy models are not engineered for the
tasks evaluated in their simulation, which is a benefit as it encourages the de-
velopment of novel approaches to the tasks evaluated by the simulation. The
simulation is sound in the sense that it appears to function consistent with the
authors stated intent, although as previously stated by the authors, there are
not many models to evaluate on their tasks. The most compelling evidence
that this benchmark is indeed beneficial in the evaluation of NeSy agents is
that the performance of the approaches evaluated by the authors falls off as the
complexity of tasks increases. This fact is evidence that the simulation poses a
legitimate challenge to researchers developing these types of agents.

**Documentation:**

The code for LogiCity is linked in the paper. Included in the zipped project
is a readme that documents how to install and run the benchmark, visualize
the simulation, evaluate pre-trained models on both tasks, train new models on
both tasks, and download required datasets. This is all the information required
to reproduce the experiment.

**Ethics:**

No issues.

**Limitations:**

1. Limitations of SPF task
The authors state they struggled to incorporate existing RL methods into their
simulation as they are carefully engineered for simpler environments. To address
this, they develop a Q-learning agent, which ultimately scores the highest in the
RL category of this task. All models seem to struggle with this task beyond the
“easy” difficulty level.

2. Limitations of VAP task
For the visual action prediction task, the authors acknowledge that there is
“...very limited literature studying FOL reasoning on RGB images...”, and so
they construct two baselines as the reasoning engine.

3 Limitations of LogiCity Simulator
The authors acknowledge that users are required to pre-define rule sets that are
conflict-free, and suggest that future work could involve ”distilling” real-world
data into LogiCity configurations. They also state that LogiCity does not sup-
port temporal logic. LogiCity is deterministic, meaning the behavior of entities
within the simulation is entirely predetermined from the initial configuration-
introducing randomness could potentially strengthen positive results.
The limitations of the proposed tasks pose an interesting conflict. The mo-
tivation of this work comes from the lack of existing configurable benchmarks
for these tasks, therefore we should not expect a wide selection of NeSy agents
to evaluate in this environment. Yet, because the authors construct these base-
lines themselves, we must trust that these baselines are created in an earnest
attempt to perform well across the various levels of difficulty available in their
benchmark. In the conclusion of their work, the authors briefly mention the
valid criticism that rules in LogiCity may not accurately reflect real life. This
is correct. And yet, the authors make the claim that LogiCity is closer to real
urban life than present alternatives.

**Opportunities For Improvement:**

The authors state that LogiCity is closer to “real urban life” than the present
alternatives. If the authors could provide some support for this claim, it would
strengthen the argument that it is a necessary advancement for the development
of more sophisticated NeSy urban agents.

**Relation To Prior Work:**

In Table 1 of the paper, the authors compare existing NeSy benchmarks to
LogiCity. Evidently, LogiCity supports various features that are not widely
available in existing benchmarks. Of the benchmarks compared against LogiC-
ity, none include all the features present in the proposed benchmark.
The primary advantage of LogiCity over existing benchmarks is its support
for customizable concepts, rules, and abstractions.

**Summary And Contributions:**

The authors present LogiCity, a simulator based on customizable first-order
logic for an urban-like environment with multiple dynamic agents. Users provide
concepts, rules, and agents which are fed into the abstraction-based simulator
to create urban grid maps. The authors include two tasks in their simulator-
Safe Path Following (SPF) and Visual Action Prediction (VAP).

---

> ### Author Rebuttal · Authors · 2024-08-16
>
> Thank you very much for your thorough review of our paper. We particularly appreciate your comment that "***the most compelling evidence...is that the performance of the approaches evaluated by the authors falls off as the complexity of tasks increases***." Flexibility and controllability are significant and unique features of our simulator, which we believe will greatly benefit the future development of Neuro-Symbolic (NeSy) methods. We address your specific comments below:
>
> > *Q1: Provide some support for “LogiCity is closer to “real urban life””*
>
> A1: Thank you for this suggestion. We offer the following qualitative evidence:
> - Compared to urban simulators like CARLA, LogiCity introduces **a much broader range of realistic behaviors and their effects**. For instance, LogiCity models "**Reckless**" and "**Ambulance**" cars. *The reckless cars drive faster when approaching intersections, increasing their chances of arriving first. When an agent encounters an Ambulance, it must stop and give way (**Appendix Table B**).* These realistic scenarios **go beyond alternatives like CARLA, which primarily models traffic lights and regular cars with simple collision-avoidance rules**.
> - Additionally, our simulator is **flexible**, it is **easily plug-and-play for new concepts and rules that emerge**, which mirrors real urban life. (For example, a new police station in the neighborhood results in the appearance of “police” cars).
>
> With these features, a NeSy agent must fully "understand" the concepts of other agents and make intelligent decisions based on such understanding. For example, if the ego agent recognizes the incoming agent is an ambulance, it should give way to it. This represents **a more sophisticated challenge than simply avoiding collisions and obeying traffic signals**.
>
> > *Q2: Limitations of LogiCity tasks and Clarification of baseline designs.*
>
> A2: Thank you for raising this point. We agree that the "limited existing approach" is not a limitation of our tasks, and we apologize for any confusion this may have caused. In designing the baselines, we have made every effort to implement and report the best results for each **under consistent criteria**. We provide **extensive implementation details in our Appendix A** and have **open-sourced all the code and pre-trained models** for these baselines to support future research.
>
> > *Q3: Are LogiCity Rules closer to real urban life than alternatives?*
>
> A3: Yes, we believe they are. To the best of our knowledge, **most autonomous driving simulators, our closest alternatives, model only a limited range of dynamic agent "concepts," primarily cars, pedestrians, and cyclists.** In contrast, LogiCity introduces a broader range of concepts and corresponding rules, such as "Reckless" cars, "Ambulances," and "Police" cars. We are planning to introduce more realistic rules and concepts in LogiCity, while they currently **have represented a significant step toward greater realism**.
>
> > *Q4: Improve writing clarity*
>
> A4: Thanks for these careful inspections, they are addressed as follows:
>
> * We have revised “prior arts” to “state-of-the-art algorithms”. We think “scenario” should be plural as in the test set, we have 100 different scenarios for each mode.
> * L314, we have revised to “configuration of the concepts”
> * L285, we have added “the”.
> * L178, we have changed to “aim to evaluate”.
> * L216, we have revised to “During testing”.
> * L263, we think “framework” should not be plural, since NLM-DQN is just one framework. Thus, we have revised the sentence to “the NeSy framework \[7, 32\] outperforms …”.

---

> > ### Comment · Reviewer_fGrq · 2024-08-20
> >
> > I am satisfied with the authors response.

---

> > > ### Author Response · Authors · 2024-08-20
> > > **Thanks for your kind response**
> > >
> > > Thank you for your thorough review of our paper and careful inspection of our rebuttal. We're glad that our responses addressed your concerns.
> > >
> > > If there are any additional aspects that could influence your final rating or if further clarification is needed, please don't hesitate to let us know.

---

### Official Review · Reviewer_p8yp · 2024-07-24
**First-order logic simulator for modelling semantic and spatial concepts in urban environments**

**Rating:** 8
**Confidence:** 4
**Correctness:** Yes.
**Clarity:** Yes.

**Review:**

Quality: The paper describes an extensive experimental analysis, particularly for safe path following, comparing the capabilities of existing inductive logical programming-based (ILP) and self-constructed neuro-symbolic (NeSy) methods against pure neural networks. The authors show that while the compared NeSy methods can efficiently learn abstractions, they struggle with complex scenarios involving numerous concepts and rules. They examined two tracks:

(i) Behaviour cloning: ILP-based algorithms, including ILP-based and NeSy ones, which were evaluated against baselines using MLPs and GNNs.
(ii) Reinforcement learning. The integration of existing NeSy methods into LogiCity proved challenging due to their specific design for particular environments. To bypass this problem, the authors developed a new Q-learning agent based on Neural Logic Machines.

Clarity: The explanations are clear.

Originality: The authors proposed using different levels of difficulty, determined by the number of concepts and rules, to test the methods' reliability.

Significance: The authors show the usefulness of the new simulator through two tasks, focusing on different aspects: (i) whether existing symbolic and neuro-symbolic methods are able to cope with the expectations arising in long-term reasoning, (ii) whether the methods can reliably focus on immediate predictions. Through these two tasks the paper shows that there is an increasing need for neuro-symbolic methods in particular to focus on real-world applications.

**Strengths:**

By addressing the limitations seen in many existing NeSy frameworks, the authors focus on providing appropriate levels of abstraction for the rules available via LogiCity, but also of flexibility to handle different levels of complexity from the environment.
Furthermore, LogiCity’s ability to offer more realistic multi-agent interactions by accounting for how agents change their actions in response to the ego agent adds value.
The user-configurable framework provides a varied environment and two useful tasks that can serve as valuable benchmarks for testing future methods.

**Additional Feedback:**

Overall, this is a well-written paper that introduces a new simulator featuring two tasks for testing models in an urban environment, with customizable difficulty levels.

**Documentation:**

Yes.

**Ethics:**

No.

**Limitations:**

Yes.

**Opportunities For Improvement:**

As methods evolve, the need for increasingly realistic datasets and benchmarks grows.
Are you considering expanding the current set (of four) possible actions to include other actions (e.g., overtaking another vehicle)?

Additionally, scenarios currently involve streets with one incoming and one outgoing lane relative to the ego vehicle. Expanding to multi-lane scenarios could include actions such as moving to the right or left lane and would add another level of realism to the simulator.
In this context, a recent paper [1] introduced the ROAD-R dataset, which includes a relevant set of actions and associated rules. This could also be useful for adding new predicates.

A minor remark: in line 263, it is stated that "NeSy framework [7, 34] outperform pure neural agents [34, 36]", and this is confusing for the reader. Perhaps, NeSy framework could be augmented by "NLM-DQN [7, 34]".

References
[1] E. Giunchiglia, M. C. Stoian, S. Khan, F. Cuzzolin, T. Lukasiewicz. ROAD-R: The Autonomous Driving Dataset with Logical Requirements. Machine Learning, 112, 3261–3291 (2023).

**Relation To Prior Work:**

Yes.

**Summary And Contributions:**

LogiCity is a first-order logic (FOL) simulator designed in the context of urban environments, capable of modelling various semantic and spatial concepts using FOL. It has two tasks: (i) safe path following for long-horizon sequential decision-making, and (ii) visual action prediction for one-step visual reasoning.

---

> ### Author Rebuttal · Authors · 2024-08-16
>
> We sincerely appreciate your valuable feedback and comments. Thank you for the positive comments: ***extensive experimental analysis, addressing the limitations seen in many existing NeSy frameworks, providing appropriate levels of abstraction, offering more realistic multi-agent interactions, and well-written paper***. We address your specific concerns and suggestions below:
>
> > *Q1: Are you considering expanding the current set of possible actions?*
>
> A1: Yes, we plan to introduce additional actions for the agents, including lane changes and specific "affordances." For instance, **a "bus" could also "pick up" pedestrians**. While current methods already struggle with our setup for **long-horizon reasoning** with abstract rules (none of the existing methods exceed 25% (trajectory) success rate in the expert mode of SPF), we can easily extend LogiCity to even more challenging tasks.
>
> > *Q2: Expanding to multi-lane scenarios could add another level of realism.*
>
> A2: Thank you for the suggestion. Given LogiCity's **modular and extensible design, incorporating real urban maps with multiple lanes is feasible**. In the initial release, we have mainly focused on the interactions at intersections and with different types of pedestrians, where the single lane was the first step, however, multi-lane is easy to add.
>
> > *Q3: Discussion of ROAD-R dataset*
>
> A3: We appreciate your recommendation. We agree that the ROAD-R dataset could offer valuable insights for designing more realistic predicates and rules. We will explore the possibility of **distilling probabilistic logical rules from the annotated atoms in ROAD-R** and integrating them into LogiCity.
>
> > *Q4: Improve Clarity of line 263*
>
> A4: Thank you for your careful review. We will revise the text to: “NLM-DQN \[7, 32\] outperforms pure neural agents \[34, 36\].”
>
> Finally, we also provide a general response to further explain our work and provide more information that you may be interested (see above). Please feel free to continue discussing with us during the reviewer-author discussion phase.

---

> > ### Comment · Reviewer_p8yp · 2024-09-01
> >
> > I am happy with the authors' response. I will keep my score.

---

### Author Rebuttal · Authors · 2024-08-16

# Overall Response

We sincerely thank all the reviewers for their valuable time and thoughtful feedback on our paper. We are pleased to see that **most reviewers have responded positively** to LogiCity. In the response, we would like to summarize the merits acknowledged by the reviewers and address the overall concerns.

## Paper presentation
We are pleased to note that the reviewers commented, ***the paper offers clear explanations*** (p8yp, x36W), ***detailed descriptions, well-designed figures*** (x36W), ***adequate writing quality, and satisfactory conceptual clarity*** (fGrq). Additionally, it is recognized for being ***logically structured, well-written, and effective*** (x36W), as well as ***clear and well-organized*** (CmLB). We have also revised the manuscript following the reviewers' detailed suggestions.
## Novelty and Significance
The main contribution and novelty of our work are:

* We introduce LogiCity, a pioneering **customizable first-order-logic (FOL)** based simulator to evaluate and advance the Neuro-Symbolic (NeSy) AI approaches.
* LogiCity explores “**compositional generalization**” challenges by introducing a **broad range of abstract concepts** and **realistic multi-agent interaction** in an urban-like environment.
* LogiCity introduced a **long-horizon reasoning task with multiple dynamic agents** (SPF) and an **abstract-reasoning task with RGB images** (VAP).

We are pleased that the benchmark’s novelty is confirmed by most reviewers: LogiCity is a ***promising*** ***new and potentially pivotal simulation for NeSy*** (fGrq, x36W, CmLB), *presents a **flexible yet challenging environment*** (p8yp, fGrq, CmLB) with ***realistic multi-agent interaction*** (p8yp, fGrq, x36W). Also, the paper is ***original*** (CmLB), has ***valuable contributions, technical innovation*** (x36W), and ***address significant gaps*** (x36W, CmLB).

As suggested by Reviewers p8yp, x36W, and CmLB, enhancing the scalability and realism of LogiCity is a crucial direction as methods continue to evolve.
1. We fully agree with this and are committed to making LogiCity more scalable and closer to real-world. We are extending LogiCity by (1) introducing more lanes, (2) expanding action spaces, and (3) introducing a probabilistic version.
2. However, we would like to emphasize that **LogiCity is already a pioneering benchmark with customizable FOL(abstraction)-based simulation**. Though it is not visually realistic, it uniquely features **realistic interactions between ego agent and multiple environment agents** with **a broad range of concepts**, which is not addressed by most existing benchmarks. In comparison to autonomous driving (AD) simulators, LogiCity models a greater diversity of concepts, such as **reckless drivers and ambulances, and their impact on the world and ego decisions**. Additionally, LogiCity’s potential impact extends **beyond AD (usually involves short-term decisions) to other areas**, including logical reasoning, symbolic planning, and reinforcement learning.
3. The primary focus of LogiCity tasks is on '**abstract reasoning**', presenting a unique and insightful challenge in the real-world. Specifically, these tasks require models to **generalize compositionally** in environments with multiple dynamic agents, a challenge that **existing RL/BC algorithms have not yet adequately addressed**.
4. Due to its innovative design and reasoning complexity, LogiCity has already posed significant challenges to most existing algorithms. E.g., while GPT-4 can answer driving-related questions with nearly 100% accuracy \[1\], it struggles with the complexities presented by LogiCity (see next section). In other words, the current LogiCity benchmark is **already highly challenging in terms of reasoning**, which urges the development of new methods.

Therefore, we fully agree with the importance of ongoing development and extension and are also encouraged that reviewers think **LogiCity is already innovative** (fGrq, x36W, CmLB), **insightful** (x36W, CmLB)**, and challenging** (p8yp, fGrq, CmLB), which are essential for an impactful benchmark and simulator.

## Experiments

To validate our benchmark, we conducted extensive experiments across various baselines to highlight the superiority of NeSy approaches and **emphasize the** **universal challenge of learning complex abstractions**.

It is encouraging to have reviewers’ comments that the proposed benchmark has ***extensive experimental analysis*** (p8yp), ***diverse urban environments, compelling evidence, sophisticated reasoning scenarios*** (fGrq). Also, the paper has ***thorough experiments, meaningful insights, comprehensive view of current capabilities*** (x36W), and the empirical results ***cover various methods and highlight key challenges*** (CmLB).
### Rebuttal Period Additions
1. **Failure Case Analysis**: Originally in **Appendix G**, we also present an in-depth analysis of **failure cases in Figure 1 of the Rebuttal PDF**. These show that SPF failures mainly arise from compositional out-of-distribution scenes, while VAP algorithms struggle with distinguishing between different reasons (concepts) for action. Video visualizations of these cases are available on our project website: https://jaraxxus-me.github.io/LogiCity/
2. **Human Performance Baseline:** We've added a human performance baseline for the VAP task on hard mode, showing that **humans outperform GPT and domain-specific models in complex reasoning tasks**. This highlights LogiCity’s potential to **drive advancements in AI models capable of abstract reasoning like humans**.
| Action/Method | Slow | Normal | Fast | Stop | Overall |
| :---: | :---: | :---: | :---: | :---: | :---: |
| **Human** | **95.0** | **92.9** | **48.0** | **83.3** | **81.2** |
| GPT4 | 40.0 | 54.8 | **28.0** | 30.0 | 40.2 |
| GPT3.5 | **55.0** | 26.2 | 16.0 | 3.3 | 23.1 |
| NLM | 40.0 | **69.1** | 0.0 | **50.0** | **44.4** |
| GNN | 37.3 | 42.5 | 12.5 | 44.4 | 38.5 |

---

> ### Author Rebuttal · Authors · 2024-08-16
>
> 3. **Significance Tests:** We provide significance tests in representative settings (**Rebuttal PDF, Tables 1, 2**), demonstrating that neuro-symbolic methods significantly outperform purely neural methods.
> 4. **Ablation Studies**: We’ve included ablation studies in the main paper on “flexible rule complexity” (**Main paper Table 1**\) and “abstraction” (**Main Paper Figure 3**). Additional ablations on visual rendering are included in the **Rebuttal PDF (Table 3)**. These studies reveal that “flexibility” is the most critical factor in increasing LogiCity's difficulty.
>
> ## Summary
>
> In each of the individual rebuttals below, we address the reviewers’ remaining concerns. All comments and questions are highly important to the continual improvement of our work, and based on them we have made or will make multiple revisions to the manuscript and appendix as promised.
>
> To summarize, we are more than happy to receive four high-quality, solid, and detailed reviews this time; thank you all. Please feel free to communicate with us during the discussion period if you have any further inquiries or questions.
>
> \[1\] Chen, Long, et al. "Driving with llms: Fusing object-level vector modality for explainable autonomous driving." 2024 IEEE International Conference on Robotics and Automation (ICRA). IEEE, 2024\.

---

### Decision · Program_Chairs · 2024-09-26

**Decision:**

Accept (Poster)

**Comment:**

This paper introduces LogiCity, a new simulator and benchmark designed to advance neuro-symbolic AI research, with a particular focus on abstract reasoning and compositional generalization within urban-like environments. The simulator offers two distinct tasks: the Safe Path Following (SPF) task, which evaluates sequential decision-making capabilities, and the Visual Action Prediction (VAP) task, which focuses on one-step visual reasoning. A key strength of LogiCity is its utilization of customizable first-order logic rules, providing flexibility in generating diverse scenarios and enabling the evaluation of various aspects of NeSy AI systems. The authors have conducted a comprehensive set of experiments, showcasing the advantages of NeSy frameworks in abstract reasoning tasks. Furthermore, their work highlights the significant challenges associated with handling complex abstractions in long-horizon multi-agent scenarios and high-dimensional, imbalanced data settings.

The majority of reviewers agree that LogiCity is a valuable contribution to the field. They appreciate its novelty, flexibility, and the challenges it poses to existing NeSy AI methods. Reviewers fGrq and p8yp rate it as 8 (Clear Accept) with high confidence, while Reviewer x36W rates it as 6 (Borderline Accept) but acknowledges its potential impact. The main concerns raised are the simulator's current level of environmental complexity and the lack of in-depth analysis of failure cases and ablation studies. The authors have addressed these concerns in their rebuttal by providing additional experiments and analysis.

Although reviewer CmLB raises concerns about the novelty of the work and its comparison to existing simulators they have not responded to the rebuttal. The authors argue that these concerns are unfounded and highlight the unique features of LogiCity, such as its support for customizable FOL and its focus on abstract reasoning. I agree with the other reviewers about the novelty of the paper.  While there is room for improvement in terms of analysis and comparison to existing benchmarks, the authors have addressed many of these concerns in their rebuttal.

Overall, LogiCity is a well-executed work that addresses a significant gap in the NeSy AI benchmark landscape. Its customizable FOL-based design, focus on abstract reasoning and compositional generalization, and extensive experimental evaluation make it a valuable tool for advancing NeSy AI research. Overall, the paper's contributions and potential impact warrant acceptance, and presenting it as a poster would provide an opportunity for further discussion and engagement with the research community.


Pros:
- Novel and flexible simulator for NeSy AI research, addressing a gap in existing benchmarks.
- Focuses on important challenges like abstract reasoning and compositional generalization.
- Offers two diverse tasks (SPF and VAP) to evaluate different aspects of NeSy AI.
- Extensive experimental evaluation across various baselines and complexity levels.
- Open-source code and data, promoting reproducibility and accessibility.

Cons:
- The current environment may be too simplistic to fully capture real-world complexity.
- The initial submission lacked an in-depth analysis of failure cases, which has now been addressed in the rebuttal.
- Further ablation studies and comparisons to existing benchmarks could strengthen the paper.
- Concerns raised by Reviewer CmLB about novelty and comparison to related work, which the authors have attempted to address in their rebuttal.